evolution/palaeontology/taxonomy and systematics

phylogenetics, dinosaurs, biogeography, diversification, paleontology, vicariance

**Author for correspondence:**
Chase Doran Brownstein
e-mail: chasethedinosaur@gmail.com

# Dinosaurs from the Santonian–Campanian Atlantic coastline substantiate phylogenetic signatures of vicariance in Cretaceous North America

Chase Doran Brownstein[1,2]

[1]Stamford Museum and Nature Center, Stamford, CT, USA
[2]Department of Ecology and Evolutionary Biology, Yale University, New Haven, CT, USA

CDB, 0000-0003-4514-9565

During the Cretaceous, diversifications and turnovers affected terrestrial vertebrates experiencing the effects of global geographical change. However, the poor fossil record from the early Late Cretaceous has concealed how dinosaurs and other terrestrial vertebrates responded to these events. I describe two dinosaurs from the Santonian to Early Campanian of the obscure North American paleolandmass Appalachia. A revised look at a large, potentially novel theropod shows that it likely belongs to a new clade of tyrannosauroids solely from Appalachia. Another partial skeleton belongs to an early member of the Hadrosauridae, a highly successful clade of herbivorous dinosaurs. This skeleton is associated with the first small juvenile dinosaur specimens from the Atlantic Coastal Plain. The tyrannosauroid and hadrosaurid substantiate one of the only Late Santonian dinosaur faunas and help pinpoint the timing of important anatomical innovations in two widespread dinosaur lineages. The phylogenetic positions of the tyrannosauroid and hadrosaurid show Santonian Appalachian dinosaur faunas are comparable to coeval Eurasian ones, and the presence of clades formed only by Appalachian dinosaur taxa establishes a degree of endemism in Appalachian dinosaur assemblages attributable to episodes of vicariance.

## 1. Introduction

Dinosaurs, mammals and other terrestrial vertebrates underwent several turnovers and diversifications as the Earth experienced

major geographical changes and shifts in floras throughout the middle and Late Cretaceous (110–66.5 Ma; e.g. [1–13]). Unfortunately, the global fossil record of the Dinosauria is especially poor for this approximately 22 Myr period (100–78 Ma, e.g. [8,10,12–14]). How the evolution of dinosaurs and other terrestrial vertebrates during this time was influenced by geographical change is debated (e.g. [1,2,15–19]), as are the origins of the high dinosaur diversity observed for the latest Mesozoic (e.g. [3]). The ability of dinosaur faunas to disperse between different continents and landmasses has remained a particularly contentious issue [2,11–13,17,18]. This debate has centered around whether local endemism or broad faunal regionalization is the prevailing pattern in dinosaur biogeography (e.g. [2,11,15,17,18]).

For approximately 30 Myr in the Late Cretaceous, eastern North America was isolated as a landmass called Appalachia [16,20–24]. Fossiliferous units that track the ancient coastline of Appalachia date to the early Late Cretaceous and have the potential to provide a wealth of information about how vertebrate faunas were then changing. Dinosaur, squamate and lissamphibian fossils suggest that the vertebrate assemblages of Appalachia, although obscured by the poor fossil record of the landmass, mainly consisted of taxa that diverged from contemporaneous relatives in western North America and Asia during the Early Cretaceous (e.g. [16,20–22]).

Here, I describe dinosaurs from the Merchantville Formation of the Atlantic Coastal Plain province in Delaware and New Jersey, USA. These dinosaurs fill a major gap in the North American record corresponding to the Late Santonian and Early Campanian Stages of the Late Cretaceous, an interval when important components of the latest Mesozoic dinosaur faunas of the Northern Hemisphere seem to have evolved [22,23]. One, a large predatory theropod, further constrains the interval in which tyrannosauroids achieved large sizes and adaptations seen in the most advanced forms, like an arctometatarsalian pes. The tyrannosauroid shows several features in its pes that ally it with the bizarre *Dryptosaurus aquilunguis* from the Maastrichtian of the Atlantic Coastline [16]. A hadrosaurid, represented by a partial skeleton and cranial and postcranial material from both adults and small juveniles, provides important new information about the evolution of the shoulder girdle in this group. Along with isolated material from the Merchantville Formation, the more complete specimens illuminate the strange nature of Appalachian vertebrate faunas and provide new evidence for the importance of geographical isolation in the evolution of dinosaurs.

# 2. Material and methods

## 2.1. Abbreviations

AMNH, American Museum of Natural History, New York, New York, USA; NJSM, New Jersey State Museum, Trenton, New Jersey, USA; YPM, Yale Peabody Museum of Natural History, New Haven, Connecticut, USA; YPM VPPU, former Vertebrate Paleontology Collection of Princeton University deposited in the Yale Peabody Museum of Natural History, New Haven, Connecticut, USA.

## 2.2. Phylogenetic methodology

Previous studies have assigned the Merchantville tyrannosauroid to Tyrannosauroidea based on morphological and phylogenetic evidence [25,26]. Although I focused on assessing the relationships of this form to other tyrannosauroids, I also conducted an additional test of this assignment by adding revised codings into the latest update of the TWiG (Theropod Working Group) matrix published by Pei *et al.* [27]. The matrix included 165 total taxa coded for 853 characters.

I used a modified version of the dataset of Carr *et al.* [28], a recently published matrix that includes 32 taxa scored for 386 characters. This dataset is among the largest made for specifically assessing tyrannosauroid interrelationships and builds on previous studies of the phylogeny of this clade [29,30]. To test the phylogenetic relationships of Appalachian tyrannosauroids more thoroughly, I included the recently published intermediate-grade tyrannosauroids *Timurlengia* [10], *Suskityrannus* [13], *Jinbeisaurus* [31] and *Moros* [12]. Given the identification of several features shared by the Merchantville tyrannosauroid and *Dryptosaurus* to the exclusion of other derived eutyrannosaurians, modification of the Carr *et al.* [28] character list was necessary to reliably test the existence of a monophyletic Dryptosauridae. Four new characters relating to features shared by the metatarsi of the Merchantville tyrannosauroid and *Dryptosaurus* were added to the matrix, and taxa for which metatarsals are known were scored on the basis of personal observation and literature reports for these characters. *Jinbeisaurus*, which is poorly constrained temporally,

was included in one round of phylogenetic analysis and excluded in another. The resulting matrix is the largest one currently compiled for assessing tyrannosauroid relationships (390 characters, 36 taxa).

I excluded two theropods from the Southern Hemisphere—*Timimus* and *Santanaraptor*—considered to be tyrannosauroids in some recent analyses (e.g. [19]) from the phylogenetic matrix presented in this paper. The evidence supporting the existence of tyrannosauroids in the Southern Hemisphere is limited, and the affinities of *Timimus*, known only from an isolated femur, to any particular coelurosaur clade are a matter of debate [7,19]. Delcourt & Grillo [19] used a number of characters related to the proximal end of the femur to place *Santanaraptor* in Tyrannosauroidea. However, this region of the femur is broken, and the external bone surface is eroded in the only known specimen of *S. placidus* ([33], fig. 6.39). Although the referral of *Santanaraptor* to any particular theropod clade awaits additional description, I note that *Santanaraptor placidus* shares with noasaurid abelisauroids like *Deltadromaeus* [2,34], *Vespersaurus* [35] and *Elaphrosaurus* [36] an elongated, non-arctometatarsalian metatarsus wherein the shaft widths of metatarsals II and IV are smaller than the shaft width of metatarsal III (a potential synapomorphy of the clade; [33], fig. 6.39), and the unexpanded medial surface of the dorsal surface of the distal end of metatarsal III noted by Sayão *et al.* [37] as a tyrannosauroid feature. Several features of the tibia and astragalus noted to unite *Santanaraptor* and tyrannosauroids among theropods by Delcourt & Grillo [19], including the absence of an accessory ridge placed on the lateral surface of the cnemial crest of the tibia and a horizontal groove across the astragalar condyles, are widely distributed among coelurosaurs (see matrix in [9]). Poropat *et al.* [32] discussed the misidentification of some noasaurids as ornithomimosaurs. Given the close phylogenetic positions and hindlimb anatomy of tyrannosauroids and ornithomimosaurs (e.g. [9,12,25,26]), that similar misidentifications could occur among tyrannosauroids and noasaurids should be considered. I thus refrain from comparisons with *Santanaraptor* and await a detailed description to do so.

In order to assess the phylogenetic affinities of the Merchantville hadrosaurid, I coded this form for the matrix of Prieto-Márquez *et al.* [23], which includes all other species of Appalachian hadrosauromorph currently considered valid. These are *Lophorhothon atopus*, *Claosaurus agilis*, *Eotrachodon orientalis* and *Hadrosaurus foulkii*. This dataset includes 63 taxa coded for 273 characters. The dataset was not modified in any way past the inclusion of the Merchantville hadrosaurid. Characters were ordered following Prieto-Márquez *et al.* [23].

The matrices were entered into the command line of the phylogenetics software TNT v. 1.5 [38]. *Allosaurus* was used as an outgroup following previous studies of the tyrannosaur dataset (e.g. [10,12,13,28,31]). For the hadrosauroid dataset, *Iguanodon* was used as the outgroup following Prieto-Márquez *et al.* [23]. In each analysis, I performed an initial search with default parameters for the options of sectorial search, ratchet, tree drift and tree fuse. I subsequently subjected the most parsimonious trees (MPTs) found using the initial search to traditional bisection-reconnection branch swapping, which more fully explores each tree island. Allowing the programme to hold 100 000 trees facilitated a large-scale exploration of tree islands. The trees generated from the analyses were summarized in strict consensus topologies, with Bremer values used to assess support for particular branches. The strict consensus trees were exported to R, and time-calibration was performed using the command timePaleoPhy in the R package paleotree [39] under the equal method of Brusatte *et al.* [40].

# 3. Results

## 3.1. Geological setting

The dinosaur assemblage described in this contribution is from the Merchantville Formation, a Late Cretaceous unit that crops out along the northern Atlantic Coastal Plain (figure 1*a*). This formation, which is composed of grey and dark green micaceous clay and fine-grained sand, underlies the Campanian Woodbury Formation within the Matawan Group (e.g. [41,42]). Strontium isotope and biostratigraphic data converge on a Late Santonian to earliest Campanian age for the Merchantville Formation [42–46]. Miller *et al.* [47] gave an age range of 84.3–77.8 Ma for this unit, which corresponds to the Late Santonian and early Middle Campanian Stages of the Late Cretaceous. Outcrop thickness varies extensively for the Merchantville Formation (12–18 m), which more commonly possesses glauconite in its upper section and at its southern end and lignite, siderite and silt to the north and in its lower beds [42,47]. The vertebrate and invertebrate assemblages documented from this formation are mostly marine in origin, as is the case for most Late Cretaceous faunas from the Atlantic Coastal Plain (e.g. [42]). The Merchantville Formation has produced the fossils of mosasaurs, large marine turtles, crocodylians including *Deinosuchus*, pterosaurs, chondrichthyan and actinopterygian fish, and a variety of ammonites and bivalves [42,43,46,48].

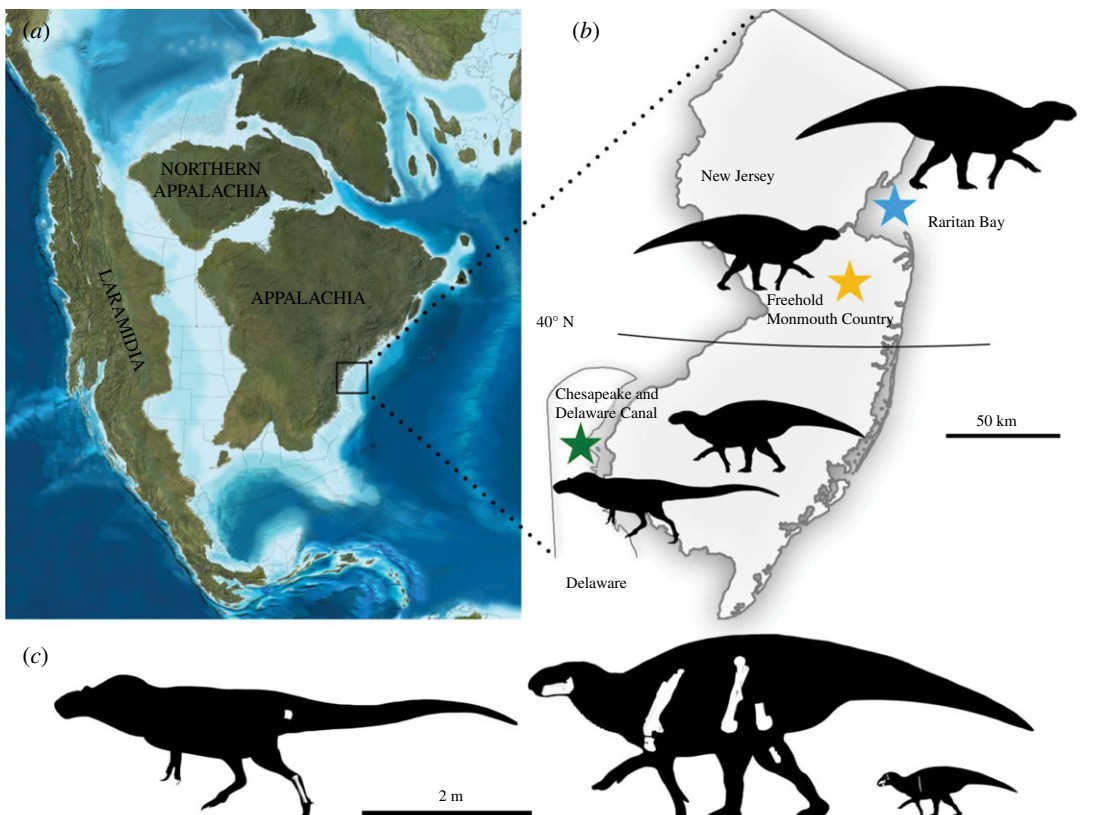

**Figure 1.** Geographic setting of the Merchantville dinosaur fauna. (*a*) Map of North American during the Campanian Stage of the Late Cretaceous (by R. Blakey, used with permission), showing the location of New Jersey and Delaware. (*b*) Map of New Jersey and Delaware showing the locations from which the described specimens were recovered. (*c*) Diagram showing preserved bones (in white) and relative sizes of (from left to right) YPM VPPU.021795 and the Merchantville hadrosaurid (adult, juvenile).

Non-avian dinosaurs previously reported from the Merchantville Formation include a large hadrosaurid metatarsal tentatively assigned to the giant hadrosaurid *Ornithotarsus immanis*, the holotype of which was recovered near Raritan Bay approximately 4 km from Keyport, New Jersey [42,49]. Gallagher [42] suggested the provenance of both of these specimens was within the Woodbury Formation based on the locality from which they were recovered. However, Gallagher [42] also correctly noted that both the Woodbury and Merchantville Formations outcrop around Raritan Bay. Clues to the origin of the holotype of *Ornithotarsus* are found in Cope [50], who noted that the specimen was recovered from clays directly underneath the lower greensand (a term for glauconite) bed and Mansfield [51], who noted that both the Woodbury and Merchantville Formations in the region around Raritan Bay were harvested for fireproofing material. An important distinction made by Mansfield [51] and other scholars (e.g. [52]) is that the Woodbury Formation is minimally glauconitic. The description given by Cope instead matches the Merchantville Formation, which is highly glauconitic at its upper part and contains siderite, silt and lignite lower in its northern exposures [42]. Based on this evidence, the Merchantville Formation is the most probable unit from which the holotype of *Ornithotarsus* originated. The new hadrosaurid material was recovered from Merchantville Formation exposures in New Jersey south of Raritan Bay approximately 8 km northwest of Freehold at the Manalapan–Marlboro township line in Monmouth County, New Jersey (figure 1*a*,*b*) [42].

The metatarsus and caudal centrum of the tyrannosauroid were recovered from exposures of the Merchantville Formation at the southern bank of the Chesapeake and Delaware (C&D) Canal approximately 1.2 km north of Summit, Delaware (figure 1*a*,*b*; [26]). At the C&D Canal, the Merchantville Formation consists of glauconitic and micaceous sand filled with silt and clay [42]. The two metatarsals were recovered in close association. The caudal vertebra was found close to the metatarsals at the same locality, and because it belongs to a large theropod of the same size as the one represented by metatarsals and shows closely comparable preservation, it most likely belongs to the

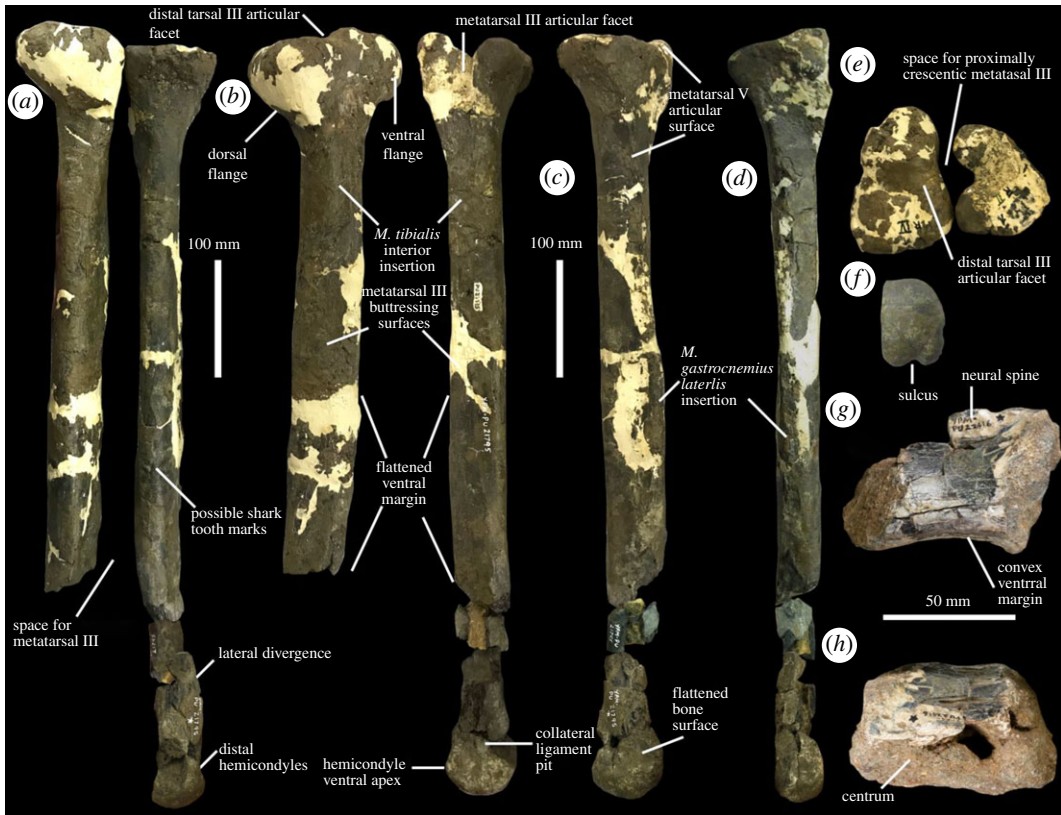

**Figure 2.** Anatomy of YPM VPPU.021795. Metatarsus in (*a*) dorsal, (*b*) medial, (*c*) lateral (metatarsal IV only), (*d*) ventral (metatarsal IV only), (*e*) proximal and (*f*) distal (metatarsal IV only) views. Caudal vertebra in (*g*) lateral and (*h*) dorsal views. Courtesy of the Division of Vertebrate Paleontology; Peabody Museum of Natural History, Yale University, New Haven, Connecticut, USA; peabody.yale.edu.

same individual. The Merchantville Formation at the C&D Canal has also preserved a diverse assemblage of vertebrate and invertebrate marine life, including turtles and mosasaurs [42].

**Systematic Paleontology.**

Theropoda Marsh, 1881

Tyrannosauroidea Osborn 1905

Dryptosauridae Marsh 1877

**Diagnosis.** Derived arctometatarsalian tyrannosauroids distinguished by the following synapomorphies among eutyrannosaurs: articular surface for metatarsal V on metatarsal IV extends distally past the proximal expansion; arctometatarsus in which metatarsal III lacks prominent diaphysial bulge that articulates with II and IV; flexor margins of metatarsals flat to concave rather than convex, such that minimal bone surface exists between the scar for the *M. gastrocnemius lateralis* and the flexor margin; no large concavity along the flexor margin of metatarsal IV proximal to the distal hemicondyles; loss of hypertrophied groove on the lateral surface of the distal end of metatarsal IV.

Dryptosauridae gen. et sp. nov.?

**Material.** YPM VPPU.021795 (holotype; figure 2*a–f*) and YPM VPPU.022416 (paratype; figure 2*g,h*), closely associated partial metatarsus and proximal caudal vertebra.

**Locality and horizon.** C&D Canal, northeastern Delaware, from the Late Santonian to Middle Campanian (approx. 84.3–77.8 Ma; [47]) Merchantville Formation. The metatarsus of YPM VPPU.021795 was found closely associated at a single site along the canal by Ralph Johnson and Ray Meyer. The similar size, colour, and preservation, close association, and individual tyrannosauroid affinities of the bones all strongly indicate they belonged to a single individual. YPM VPPU.022416 was collected from the same locality (the collection records are identical) in very close proximity to the metatarsus based on the museum record [53], and donated by Wayne Cokeley. Given that YPM VPPU.022416 preserved in a similar manner as YPM VPPU.021795 and was recovered very close by the metatarsals at the same site, these bones are probably from the same individual.

**Description.** The fragmentary skeleton includes the partial pes and caudal vertebra of a mid-sized tyrannosauroid (figure 2). This specimen is from an adult or large subadult individual based on the

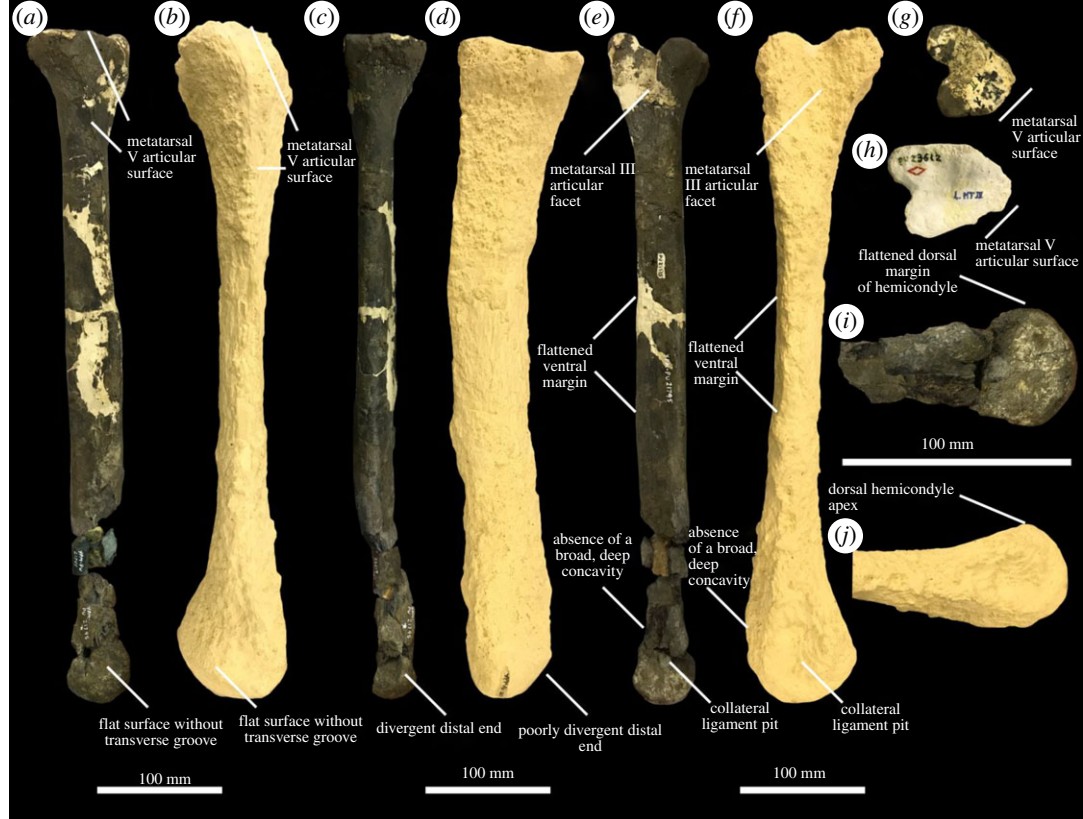

**Figure 3.** Distinguishing features of dryptosaurids. Metatarsals IV of YPM VPPU.021795 (*a,c,e,g,i*) and *Dryptosaurus aquilunguis* (*b,d,f,h,j*) in lateral (*a,b*), dorsal (*c,d*), medial (*e,f*) and proximal (*g,h*) views, with detailed photographs of the distal end of the metatarsals IV in medial (*i,j*) views. Courtesy of the Division of Vertebrate Paleontology; Peabody Museum of Natural History, Yale University, New Haven, Connecticut, USA; peabody.yale.edu.

prominence of the muscle and ligament attachment sites along the metatarsals and the fusion of the neural arch base to the centrum of the caudal vertebra. Parts of the metatarsus have been stabilized with plaster, but potentially diagnostic features of the Merchantville tyrannosauroid occur only along clearly preserved regions of the proximal ends of metatarsals II and IV as shown in figure 2*c–e*. Metatarsals II and IV were originally incorrectly labelled as the opposite element (see writing in figure 1*e*). Tyrannosauroid hindlimb material is considered diagnostic to the species level (e.g. [12,13,16]), and so there is precedent for naming a new taxon based on relatively little material from the postcranium and specifically the hindlimb. I take a conservative approach by suggesting novel apomorphies for the Merchantville tyrannosauroid and thus presenting Dryptosauridae as a multi-species clade while not erecting a new name.

The left metatarsus included in YPM VPPU.021795 (figure 2*a–d*) is elongated compared to the robust metatarsi of tyrannosaurids and other tyrannosauroids with the exceptions of *Alectrosaurus olseni* and *Moros intrepidus* (figure 3; [12,54]). The Merchantville form is allied with eutyrannosaurian tyrannosauroids among coelurosaurs based on the following features: the presence of a deep, V-shaped notch on the proximal end of metatarsal IV for articulation with a proximally crescentic metatarsal III (figure 2*e*; electronic supplementary material, figure S1), large, teardrop-shaped buttressing surfaces for metatarsal III on the medial faces of metatarsals II and IV (figure 2*b*; electronic supplementary material, figures S2 and S3), and well-developed attachment sites for the *M. gastrocnemius lateralis* along the flexor margins of each bone separated from the metatarsal III by a ridge (figure 2*d*; [12,13,16,22,28,30,53–59]). These characters, along with the presence of an arctometatarsalian metatarsus (unfused metatarsals II–IV are closely appressed and the proximal end of metatarsal III is hidden in extensor view) itself, also ally the Merchantville tyrannosauroid with Eutyrannosauria to the exclusion of other tyrannosauroid clades.

This combination of features also allows for a rejection of the hypothesis that YPM VPPU.021795 represents either a robust ornithomimid pes or the partial metatarsus of a caenagnathid (also see the phylogenetic analyses of Coelurosauria including YPM VPPU.021795 in [25], this paper). At least two genera of ornithomimids evolved metatarsals that resemble the same elements in tyrannosaurids in robusticity.

However, these species lack the deep notch on metatarsal IV for metatarsal III, possess linear, rather than clearly teardrop-shaped, buttressing surfaces, and are strongly distally divergent [60,61]. Giant caenagnathids that approached the masses of large tyrannosauroids are known from the Cretaceous of western North America [62] and Asia [63]. Arctometatarsalian caenagnathids lack a deepened articular facet on metatarsal IV for a proximally crescentic metatarsal III (e.g. [64,65]) characteristic of arctometatarsalian tyrannosauroids [28,30,54]. Arctometatarsalian caenagnathids have distally displaced, poorly developed buttressing surfaces for metatarsal III on metatarsals II and IV that abut the distal condyles (fig. 10b in [64]); these are prominent and placed near the diaphysis in the Merchantville form and other arctometatarsalian tyrannosauroids (figure 2b; electronic supplementary material, figure S3; [12,22,54,56]). The metatarsals II of arctometatarsalian caenagnathids are approximately square in proximal view (e.g. [64], fig. 3C1; [65]), whereas the Merchantville form shares with tyrannosauroids a discrete posteriorly projecting process in proximal view (figure 2e). Electronic supplementary material, figure S1 shows the metatarsal II of the Merchantville form compared to those of tyrannosauroids, ornithomimids and caenagnathids in proximal view. Finally, the metatarsal IV of the Merchantville form shows the distinct laterally 'kinked' divergence (figure 2a) found in tyrannosauroids but not caenagnathoids. This feature was previously used to distinguish arctometatarsalian tyrannosauroid metatarsals from caenagnathids by Peecook *et al.* [54].

Metatarsal II is expanded proximally (figure 2a,b), with an extremely bulbous proximal end relative to the metatarsals II of other tyrannosauroids [9,16,22,28,30,54–57,59,66]. The proximal surface of metatarsal II bears a deepened excavation for the articulation with distal tarsal III compared to the condition in other eutyrannosaurians, including *Tyrannosaurus*, *Appalachiosaurus*, *Alioramus*, *Bistahieversor*, *Gorgosaurus* and *Albertosaurus* [12,22,56,58,66,67]. This feature is not attributable to ontogenetic variability, as immature individuals within the genera *Bistahieversor* [67], *Albertosaurus* [66] and *Appalachiosaurus* [22], and indeterminate tyrannosaurids [54] lack the deep sulcus present on the metatarsal II of YPM VPPU.021795. The diagnostic utility of this feature is supported by comparison with the metatarsals II of other arctometatarsalian coelurosaurs like ornithomimosaurs, which also lack the condition in YPM VPPU.021795 (e.g. [12,60,61]).

The proximal surface of metatarsal II lacks a distinct articular facet for metatarsal III (figure 2b,d). Instead, the medial surface is flattened in proximal view (figure 2d). This is unlike the condition in all other eutyrannosaurs for which metatarsal II is known, where an articular facet forming a deep notch is present [13,16,22,28,30,54–59,68]. The condition in YPM VPPU.021795 cannot be explained by ontogeny, as the holotype of *Appalachiosaurus*, a subadult, possesses a deepened notch at the proximal end of metatarsal II [22], as do immature specimens of *Albertosaurus* [66] and *Bistahieversor* [67]. A deep facet at the proximal end of metatarsal II also appears in some ornithomimids, including *Quipalong*, *Aepyornithomimus* and *Ornithomimus velox* [60,61,69].

Just below the proximal expansion of metatarsal II, two distinctive regions of textured, emarginated bone are observable (figure 2b). One small region is identifiable as the insertion scar for the *M. tibialis interior* [54]. This insertion pit consists of smooth bone surface with several striations running through it that is emarginated by two low eminences of bone dorsally and ventrally. The larger region is teardrop-shaped and identifiable as the buttressing surface for metatarsal III. The moderately developed nature of this feature is unlike the derived condition in tyrannosaurids, wherein the buttressing surface on metatarsal II is strongly developed and far more defined (e.g. [22,30,54,56,59]). Instead, it matches the condition in *Appalachiosaurus* and stemward taxa.

Parallel to the ventral margin of metatarsal II is a linear groove identifiable as the insertion scar for the *M. gastrocnemius lateralis* [70]. The metatarsal shaft lacks a convex flexor margin and a prominent flexor–extensor diaphysial bulge unlike juvenile and adult specimens of other eutyrannosaurs [22,54,56,58,59,66], two features also seen in metatarsal IV in YPM VPPU.021795. The bone surface of this metatarsal II is poorly preserved in some areas, but the insertion scar for the *M. gastrocnemius lateralis* is still discernible towards the distal end of the shaft. The shaft of metatarsal II is straightened to very slightly laterally divergent towards its distal end of the bone.

Metatarsal IV is elongate and mediolaterally compressed. Among arctometatarsalian tyrannosauroids, only the non-tyrannosaurid tyrannosauroids *Alectrosaurus olseni* and *Moros intrepidus* possess similarly elongated metatarsals (figure 2a–d; [12,54]). However, the metatarsal IV of the Merchantville form differs substantially from that of *Moros* in relative proportions [12]. An earlier paper by the author [25] overestimated the length of this bone when complete, but the recognition of a fragment of the shaft that unites the proximal portion with the distal end allows for a revision of the robusticity index of this bone (see earlier). The deepened proximal articular facet for metatarsal III on metatarsal IV is crescentic in dorsal view. This facet would have articulated with a proximally crescentic metatarsal III (figure 2e). Below the proximal expansion, the insertion pit for the *M. tibialis interior* and the teardrop-shaped

articular surface for metatarsal III are both visible and similarly positioned as in the metatarsal II of YPM VPPU.021795. Metatarsal IV is ventrally flattened rather than convex, as in the metatarsal II included in YPM VPPU.021795, and there is no prominent concavity between the main body of the shaft and the distal hemicondyles (figure 2*a*,*b*). The insertion scar for the *M. gastrocnemius lateralis* is visible but poorly defined (figure 2*c*).

The distal portion of metatarsal IV is also preserved in YPM VPPU.021795 and includes the distal hemicondyles. Metatarsal IV is distally mediolaterally compressed, such that the flexor–extensor width of the bone is greater than the mediolateral dimension. This produces a mediolaterally compressed, subrectangular shape for the bone in distal view (figure 2*f*). In *Moros intrepidus*, metatarsal IV is similarly mediolaterally compressed, but the bone in that taxon and in virtually all other intermediate-grade tyrannosauroids and tyrannosaurids is subrectangular in distal view (e.g. [12,16,22,54,66]). The distal hemicondyles are confluent with the rest of the shaft of the metatarsal along the flexor margin, instead of being separated from it by a moderately to strongly developed concavity as in other arctometatarsalian tyrannosauroids (e.g. [12,22,54–56]). The medial collateral ligament pit is circular and deepened, whereas no lateral collateral ligament pit is observable. Instead of a clearly defined pit or groove on the lateral surface of the distal hemicondyles, smooth bone surface is present. The flexor margins of the distal hemicondyles converge to sharp apices rather than sporting broadly concave outlines.

The caudal vertebra YPM VPPU.022416 (figure 2*g*,*h*) closely resembles the caudal vertebrae of *Dryptosaurus* and other large-bodied tyrannosauroids [16,53,67]. The ventral margin of this bone is caudally expanded and strongly concave, unlike the caudal vertebrae of caenagnathids (e.g. [71]). Although neither the cranial nor the caudal edge of this bone was preserved, its general morphology indicates it was slightly opisthocoelous. The neural arch and spine were moderately developed, as indicated by the preserved portion of the neural arch (figure 2*e*). Taken together with the size of the vertebra relative to the metatarsus YPM VPPU.021795, the presence of a developed neural arch and spine suggests this vertebra is a proximal caudal (*sensu* [53]).

**Comparisons with *Dryptosaurus.*** Several important features are shared in the metatarsals of the holotypes of *Dryptosaurus* and the Merchantville tyrannosauroid (figure 3*a–j*). These include an enlarged articular surface for metatarsal V on metatarsal IV that terminates distal the expanded proximal head (figure 3*a*,*b*,*g*,*h*) rather than above the proximal expansion as in *Suskityrannus, Appalachiosaurus, Alectrosaurus* and tyrannosaurids [12,13,16,22,54–57,59,72]. The shafts of the metatarsals II and IV of the Merchantville tyrannosauroid and *Dryptosaurus* also both lack convex ventral margins that create a surface of bone positioned ventrally relative to the *M. gastrocnemius lateralis* insertion groove (figure 3*a*,*b*,*e*,*f*). In other arctometatarsalian tyrannosauroids, this margin is clearly developed, creating a broadly to acutely ventrally convex profile for metatarsals II and IV (e.g. [12,16,22,54–56]). The metatarsals of the Merchantville tyrannosauroid and *Dryptosaurus* both show a heavily reduced to absent ventral concavity just proximal to the distal hemicondyles, a feature that is heavily developed in other tyrannosaurs [12,16,22,54–57,59]. The presence of a convex ventral margin just preceding the distal hemicondyles in metatarsal IV is significant for its use in justifying the assignment of some specimens to Tyrannosauroidea over other theropod clades [12]. Finally, *Dryptosaurus* and the Merchantville tyrannosauroid are distinguished from other arctometatarsalian tyrannosauroids by the absence of a distinctive groove or pit along the lateral surface of the distal hemicondyles of metatarsal IV. In other arctometatarsalian tyrannosauroids, a groove is present in this area (e.g. [12,16,22,54,56]).

One other important feature in the metatarsals of *Dryptosaurus* and the Merchantville form that distinguishes them from those of other eutyrannosaurs concerns the construction of the arctometatarsus. In tyrannosaurids and other arctometatarsalian tyrannosauroids, metatarsal III bulges towards its distal end before abruptly thinning and disappearing from dorsal view distal to the proximal end of the metatarsus [12,13,16,22,54–57,59]. This condition is also present in some ornithomimids (e.g. [60,61,69,73,74]). *Dryptosaurus* and the Merchantville tyrannosauroid possess a different configuration wherein metatarsal III lacks developed diaphysial bulge, maintains its mediolateral width in dorsal view throughout its proximodistal run, and is only dorsally obscured at the proximal end of the metatarsus. This morphology is present in the Merchantville tyrannosauroid based on the shape of the space for metatarsal III and the proximal extent and width of the buttressing surfaces on metatarsals II and IV (figure 2*a*,*b*) and is displayed by the preserved portion of metatarsal III in the holotype of *Dryptosaurus* ([16], fig. 21A,B), the morphology of which is slightly reinterpreted here. Although I concur with the identification made by Brusatte *et al.* [16] of one bone as metatarsal III, I reinterpret the fossa identified by Brusatte *et al.* [16] as the fossa located proximal to the distal hemicondyles on the ventral surface. The ridge mentioned by Brusatte *et al.* [16] is most likely the ventral ridge that forms the border of the articular surfaces for metatarsals II and IV on metatarsal III and is located *proximal* to the fossa. These

features are commonly found on the ventral surfaces of tyrannosauroid metatarsals III (e.g. [22], fig. 19 k; [66], fig. 15c), and the metatarsal III of *Dryptosaurus* was likely flipped along its dorsoventral and proximodistal axes by Brusatte *et al.* [16]. The metatarsal III of *Dryptosaurus* lacks a mediolaterally divergent diaphysial bulge in both my interpretation and that presented in Brusatte *et al.* [16]. The distal ends of the metatarsals II and IV of *Dryptosaurus* and the Merchantville tyrannosauroid do not sharply diverge outward to accommodate the expansion of metatarsal III, a feature that demonstrates the diaphysial bulge of metatarsal III was reduced in the Merchantville tyrannosauroid as it is in *Dryptosaurus* (see earlier; figure 3c,d). As in other tyrannosauroids with the arctometatarsalian condition, the proximal end of metatarsal III was obscured dorsally by the proximal ends of metatarsals II and IV in the Merchantville tyrannosauroid and *Dryptosaurus*.

The Merchantville tyrannosauroid differs from *Dryptosaurus* in its far more gracile metatarsal IV that has a triangular, rather than subrectangular, outline in proximal view (figure 3g,h). *Dryptosaurus* also differs from the Merchantville tyrannosauroid and other arctometatarsalian tyrannosauroids in the morphology of the distal hemicondyles of metatarsal IV. The extensor margin of the distal hemicondyles of the metatarsal IV in *D. aquilunguis* forms a prominent apex, whereas in other tyrannosauroids this region is flat and confluent with the extensor margin of the bone shaft (figure 3i,j; electronic supplementary material, figure S2A–F; [12,22,54–56,59,67]). The distal hemicondyles of *Dryptosaurus* are also triangular in outline in distal view and separated by a shallow, broad sulcus (fig. 22f in [16]). By contrast, the metatarsal IV of the Merchantville tyrannosauroid is rectangular and mediolaterally compressed in distal view, and the hemicondyles are separated by deepened sulcus (figure 2f). The mediolateral compression of the metatarsal IV of the Merchantville form in distal view is shared with *Moros intrepidus* among arctometatarsalian tyrannosauroids [12].

**Systematic Paleontology.**

Ornithischia Seeley 1887

Hadrosauromorpha Norman 2014

Hadrosauridae Cope 1869

Hadrosauridae gen. et. sp. nov.?

**Material.** YPM VPPU.021813, partial skeleton of a single adult including both coracoids, both scapulae, a femur, a fragmentary proximal tibia and several fragments. Associated cranial and postcranial material from multiple perinatal individuals that includes a quadrate, several partial maxilla portions, a partial jugal, skull roof fragments and several rib fragments is also included in specimen YPM VPPU.021813. AMNH 7626 (YPM VPPU.021824, cast), dentary. YPM VPPU.04508, rib, femur and long bone portions. AMNH 13704, partial dentary of a probable perinate hadrosaurid recovered from the same location as the holotype and referred material [75]. The adult specimen included in YPM VPPU.021813 and the additional bone portions in YPM VPPU.04508 may belong to the same individual based on their extremely similar size, matching preservation and closely similar colour, but out of an abundance of caution YPM VPPU.04508 is only tentatively referred to the same form.

**Locality and Horizon.** Approximately 8 km northwest of Freehold near Manalapan–Marlboro township line, Monmouth County, New Jersey; Late Santonian to Middle Campanian (approx. 84.3–77.8 Ma) [47] Merchantville Formation. The scapulae, coracoids and hindlimb material in the paratype are from the same individual given their matching sizes, the lack of duplicated bone elements, the similar excellent preservation of the elements and association. YPM VPPU.04508 consists of a number of rib, femur and long bone fragments recovered from the same site. These bones are likely from the same individual given the lack of overlapping material (the femur fragments in YPM VPPU.04508 are identified as being from the right femur). Given that all of these bones were recovered from the same site, and only one adult and one juvenile individual are confidently represented, I tentatively refer all of these specimens to the same taxon of hadrosaurid. In the western USA, monospecific hadrosaurid sites are known wherein both small juvenile and large adult individuals are preserved (e.g. [76,77]). I tentatively interpret the Manalapan–Marlboro site as another example of this type of association.

**Description.** The new adult Merchantville hadrosaurid cranial (figure 4a,b) and postcranial (figures 5 and 6) material are identifiable as such based on the presence of a dentary occlusal plane parallel to the lateral surface of the dentary ramus, a long, hooked ventral process on the coracoid, an elongate scapula with a developed acromion process and a deepened deltoid fossa, and a femur with a poorly separated lesser trochanter (e.g. [23,24,78,79]). Small skull and postcranial bones recovered from the same site as the dentary and postcranial skeleton may be assignable to the same taxon represented by the ontogenetically mature specimens based on their identical phylogenetic placement when the adult postcrania and juvenile cranial material are coded as individual OTUs, their close association and similar preservation to the adult material, and their individual assignability to basal hadrosaurid dinosaurs. The partial

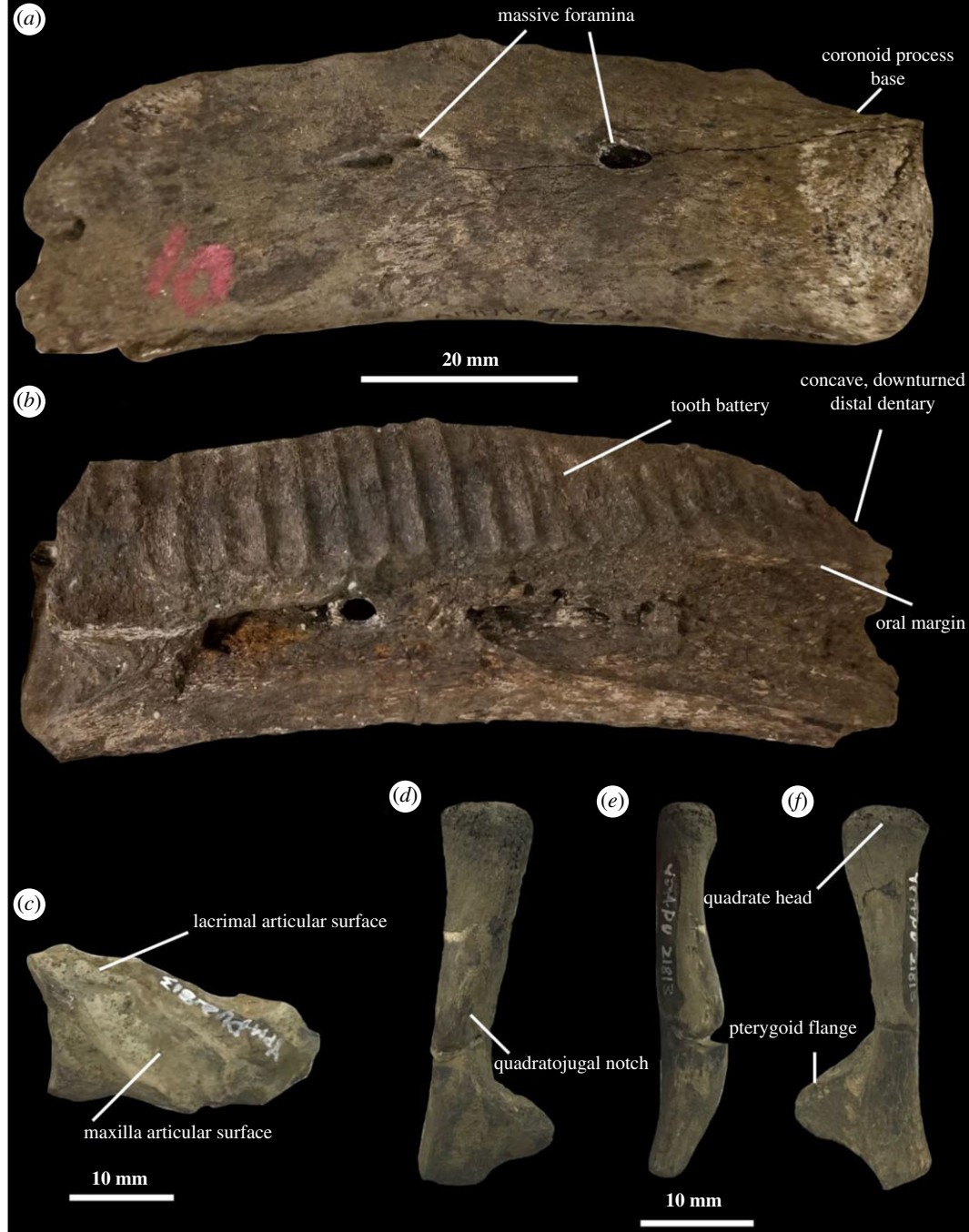

**Figure 4.** Cranial material of the Merchantville hadrosaurid material AMNH 7626 and fragments included in YPM VPPU.021813. Dentary AMNH 7626 in lateral (*a*) and medial (*b*) views. (*c*) Partial jugal included in YPM VPPU.021813 in lateral view. Quadrate included in YPM VPPU.021813 in (*d*) lateral, (*e*) proximal and (*f*) medial views. Courtesy of the Division of Vertebrate Paleontology; Peabody Museum of Natural History, Yale University, New Haven, Connecticut, USA; peabody.yale.edu.

postcranium and dentary likely come from individuals nearing or at osteological maturity, given the smooth surface of the scapula and femur, the large number of tooth positions in the dentary, and the presence of rugose muscle attachment sites on the scapula and the fourth trochanter of the femur. Although a number of fragments exist of this portion of the skeleton, only those that can be confidently identified are described here. For the sake of organization, I describe the skull material referred to the potentially novel Merchantville hadrosaurid before discussing the holotype postcrania.

The preserved juvenile cranial material in YPM VPPU.021813 includes a partial jugal (figure 4*c*), a quadrate (figure 4*d–f*), a partial dentary described by Gallagher [75], as well as possible maxilla, skull roof and rib material. The partial jugal includes the articular surface of this bone (figure 4*e*). The jugal

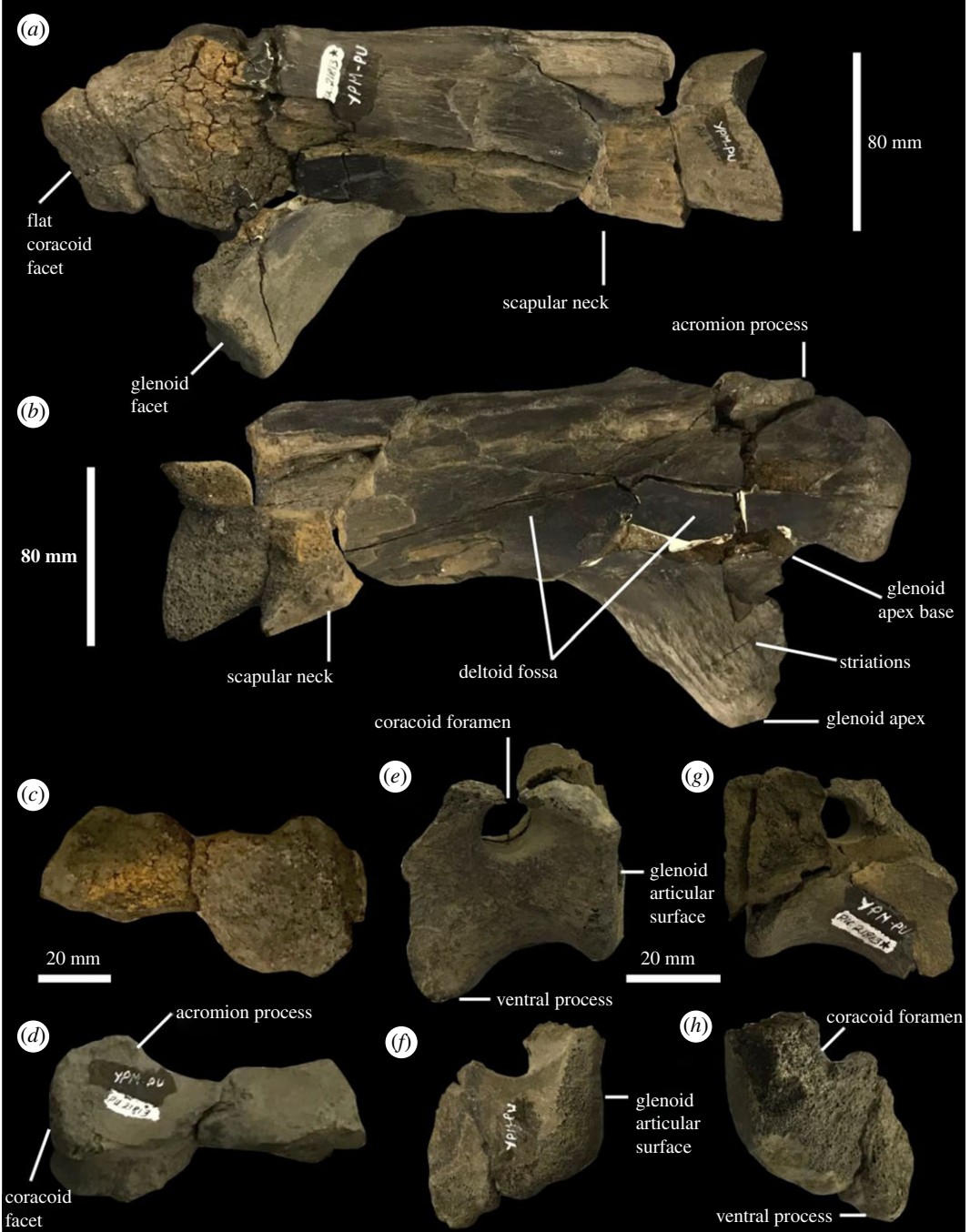

**Figure 5.** Shoulder girdle of YPM VPPU.021813 (adult). Right scapula in (*a*) lateral and (*b*) medial views. Proximal left scapula in (*c*) lateral and (*d*) medial views. Left and right coracoids in (*e,f*) lateral and (*g,h*) medial views. Courtesy of the Division of Vertebrate Paleontology; Peabody Museum of Natural History, Yale University, New Haven, Connecticut, USA; peabody.yale.edu.

process apex is present and moderately developed, and the apex of this feature is dorsally positioned. A small eminence positioned on the dorsal margin of the jugal is identifiable as the lacrimal process [24], although the anatomy of this feature cannot be discerned. The articular facet for the maxilla is large and widened, whereas the articular surface for the lacrimal is smaller and dorsally positioned. The rostral process of the jugal in the juvenile Merchantville hadrosaurid is slightly constricted towards its caudal end, as in *Eotrachodon* [23,24], saurolophines including all four species of *Gryposaurus* [80], *Brachylophosaurus* and *Probrachylophosaurus* [81], *Saurolophus* [82] and *Edmontosaurus regalis* [83], and lambeosaurines including *Olorotitan* [84], *Amurosaurus* [85], *Velafrons* [86] and *Magnapaulia* [87], but unlike the hadrosauromorph *Gilmoreosaurus* [88].

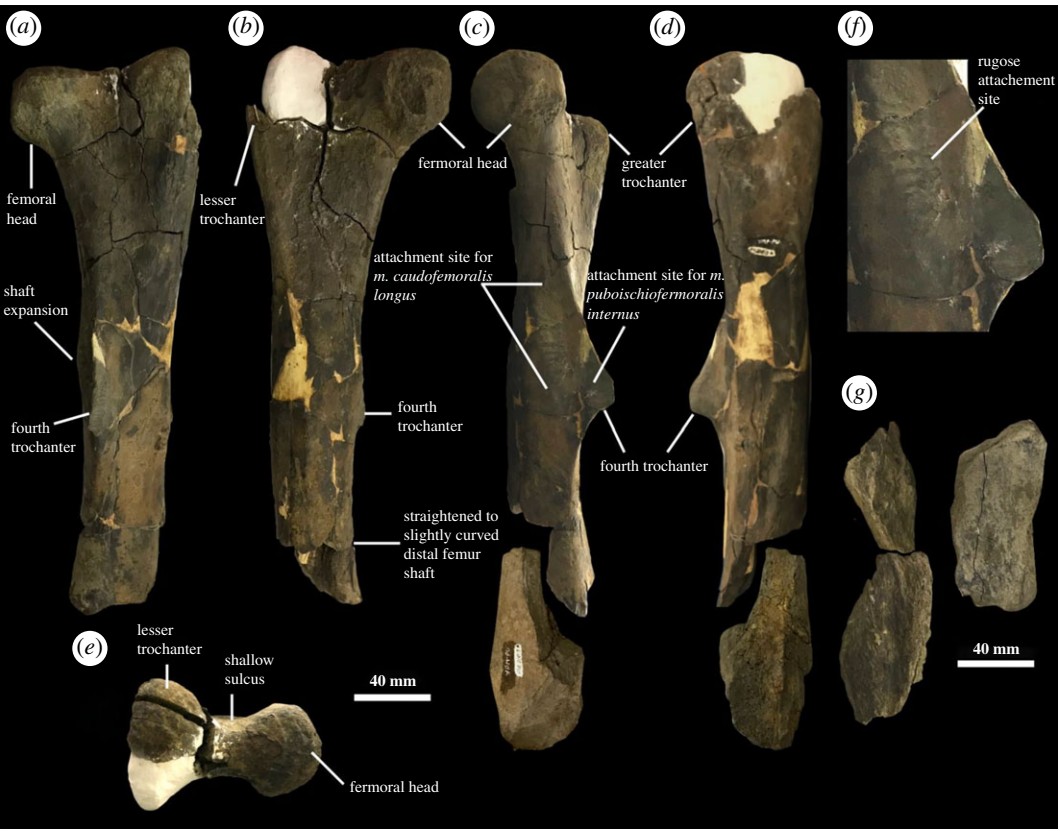

**Figure 6.** Hindlimb material of YPM VPPU.021813. Left femur in (*a*) ventral, (*b*) dorsal, (*c*) medial, (*d*) lateral, and (*e*) proximal views, with a detailed photograph of muscle attachment surfaces near the fourth trochanter (*f*). Fragmentary femur in YPM VPPU.021824 (*g*). Courtesy of the Division of Vertebrate Paleontology; Peabody Museum of Natural History, Yale University, New Haven, Connecticut, USA; peabody.yale.edu.

The columnar quadrate is elongate and gracile (figure 4*d–f*) like those from perinate individuals of *Maiasaura* [77]. Compared to *Eotrachodon* [24], *Lophorhothon* [89], and skulls of *Brachylophosaurus, Gryposaurus, Edmontosaurus, Probrachylophosaurus, Saurolophus, Magnapaulia, Amurosaurus* and other derived hadrosaurid genera, this bone possesses larger lateral and medial flanges along the quadrate process and a weakly curved quadrate shaft (e.g. [80–83,85–87]). The quadrate head is convex, and the proximal quadrate lacks a buttressing surface for the squamosal along its caudal surface. The lateral flange of the quadrate is poorly developed. The pterygoid flange is heavily developed and dorsoventrally constrained to produce a 'pinched' morphology in medial view. This flange is triangular in shape and sharply diverges from the main quadrate body. The quadrate is depressed along its medial surface to articulate with the pterygoid. The quadratojugal notch is gently concave and arcuate. This notch begins to develop near the midpoint of the quadrate, indicating the midpoint of the notch was ventrally offset.

The Merchantville hadrosaurid dentary AMNH 7626 is deep and lacks the condition seen in *Eolambia* [90], *Protohadros* [91], *Lophorhothon* (electronic supplementary material, figure S4C), lambeosaurines [84,85] and saurolophines [80,82,83] of a strongly distally ventrally concave lateral profile (figure 4*a,b*; electronic supplementary material, figure S5). Instead, the dentary is broadly concave along its ventral margin, a condition that matches the morphology of the dentaries of basal hadrosaurids like *Eotrachodon orientalis* [24] and *Aquilarhinus palimentus* [79] and hadrosauromorphs like *Gobihadros* [92], *Gilmoreosaurus* (electronic supplementary material, figure S4B; [88]) and *Bactrosaurus* [93]. The distal end of the dorsal margin of the dentary AMNH 7626 also does not converge to a particular point and suddenly diverge ventrally as in *Eotrachodon* [24]. At least two distinct rows of ovoid foramina are present (figure 4*a*). The dorsal of the two includes an unusually massive, deepened foramen towards its proximal end that communicates with a distinctive foramen in the interior of the dentary. This foramen broadly communicates with the internal neurovasculature, as damage to the medial surface of the dentary reveals. The foramina rows on the dentary AMNH 7626 are sparsely populated relative to the condition in *Eotrachodon*, wherein a distinctive row of numerous foramina is present along the

lateral surface of the dentary [24]. The coronoid process base is preserved and shows this feature was dorsally and slightly caudally directed. The juvenile dentary fragment is essentially identical to the adult dentary in morphology [75].

As in hadrosaurids but unlike basal hadrosauroids, the occlusal plane does not strongly diverge from the lateral margin of the dentary ramus, and the alveolar row is nearly straight in dorsal view [23,24,79]. There are 23 (est. 30+) preserved tooth positions in the partial dentary AMNH 7626, a higher estimated number than in the juvenile holotype of *Eotrachodon orientalis* [24]. Unlike other basal hadrosaurids and hadrosauromorphs, the ventral margin of the tooth battery is not broadly ventrally convex in lateral view, and instead is flattened to concave in shape [24,79,90–92,94]. The inside of the dentary is perforated by a canal for the insertion of the neurovasculature of the lower jaw, including the inferior alveolar nerve.

The shoulder girdle of the adult skeleton in YPM VPPU.021813 is among the most complete known for any Appalachian hadrosauromorph (figure 5; [23,24,49,89]). The coracoids possess widened, ovoid, and caudally placed coracoid foramina and subtly developed biceps tubercles (figure 5*e–h*). The apices of the ventral processes of the coracoids are only partially preserved. These processes were clearly developed and hooked as in derived hadrosauromorphs like *Lophorhothon* [95] and hadrosaurids like *Hadrosaurus*, *Edmontosaurus*, *Magnapaulia*, *Olorotitan* and *Amurosaurus* [24,49,84,85,87,96] but unlike the reduced ventral processes present in *Gilmoreosaurus* [88], *Gobihadros* [92], *Eolambia* [90] and *Bactrosaurus* [93]. The glenoid articular surfaces of the coracoids are flattened.

Both of the scapulae are represented in the adult skeleton included in YPM VPPU.021813 (figure 5*a–d*). The coracoid facet is far smaller than the glenoid facet and dorsally positioned along the proximal margin of the scapular head. The heavily developed glenoid facet culminates in the glenoid apex, which appears as a strongly developed, ventrally divergent subtriangular process sporting a striated bone surface (figure 5*b*). In YPM VPPU.021813, the development of the glenoid process as measured from its base to the glenoid apex is high, contributing to more than half of the total depth of the anterior scapula below the acromion (figure 5*b*; ratio 61.5/108.7 = 0.56577). The same feature was used by Párraga & Prieto-Márquez [97] to differentiate the rhabdodontid *Pareisactus evrostos* from other members of the Iguanodontia, which includes the clade Hadrosauromorpha. Although they are not considered autapomorphic here given the variability of attachment sutures, the prominent striations surrounding the glenoid apex in YPM VPPU.021813 have not been reported in multiple juveniles or adults of other hadrosauromorphs [79,84,85,87,93,96,98]. The deltoid fossa of the scapula is deepened and underlies a poorly developed pseudoacromion process that projects dorsolaterally. The scapular blade is broken and little more can be said about its morphology besides that it was expanded.

The hindlimb in YPM VPPU.021813 is represented by a mostly complete femur (figure 6*a–d*) and a fragment of the tibia. The femur is elongate and straightened to very slightly bowed, as in derived hadrosauromorphs and hadrosaurids [23,24,49,79,88,92,94] but unlike basal hadrosauromorphs (e.g. [90]). The proximal end of the femur preserves three distinct features: the femoral head, the greater trochanter and the lesser trochanter. The femoral head projects outward medially and is rectangular in dorsal and ventral views. The femoral head is not bowed from the femoral shaft as in basal hadrosauromorphs like *Eolambia* [90] and is also far more laterally divergent than those of the femora of lambeosaurines (e.g. [79,84,85]). The lesser trochanter is poorly developed and confluent with the greater trochanter.

Halfway along the proximodistal run of the femur in YPM VPPU.021813, the fourth trochanter is observable. A distinctive line of rugose bone surface formed by developed striations lines the fourth trochanter, which is ventrally smooth and lacks a sharp apex. This rugose bone surface represents the attachment site for the *M. caudofemoralis longus* (figure 6*f*; [99]). The attachment site for the *M. puboischiofemoralis internus* appears as a smooth, ovoid fossa positioned at the medial–proximal corner of the bone surface surrounding the fourth trochanter (figure 6*f*; [99]). The fourth trochanter converges to a ventrally directed point along its proximodistal run; this is the plesiomorphic condition for hadrosauromorphs (e.g. [23,79,99]). This represents a character state reversal, as non-hadrosaurid hadrosauromorphs close to the root of Hadrosauridae like *Gilmoreosaurus* [88], *Bactrosaurus* [93] and *Gobihadros* [92] all show the smooth condition present in hadrosaurids. The femur in YPM VPPU.021813 displays the highest value among basal hadrosauromorphs and hadrosaurids for the ratio of the mediolateral shaft width below the proximal head to the mediolateral shaft width below the fourth trochanter (figure 7). A high value for this ratio was used to distinguish the European hadrosaurid *Pararhabdodon* from other hadrosauromorphs [100].

Only a small probable portion of the tibia was preserved, potentially representing a fragment of the shaft below the proximal expansion. A small groove may correspond to the intercondylar sulcus.

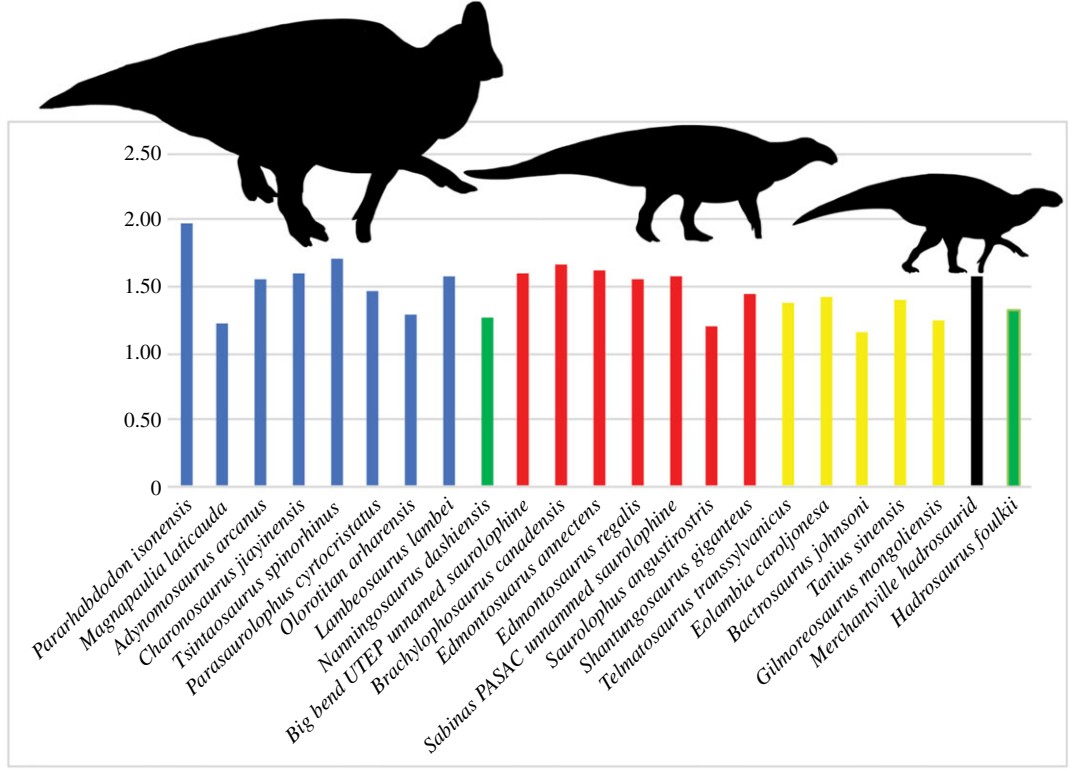

**Figure 7.** Femoral constriction in hadrosauromorphs. The y-axis indicates the ratio between the depth of the glenoid process and the total depth of the anterior scapula. Blue bars denote lambeosaurines, green denote basal hadrosaurids, red denote saurolophines, yellow denote hadrosauromorphs, and black denotes the Merchantville hadrosaurid YPM VPPU.021813.

In addition to these materials, long bone fragments are included in the set YPM VPPU.04508. These clearly came from large bones comparable in size to the femur and tibia in YPM VPPU.021813. I tentatively identify several fragments included in YPM VPPU.04508 as additional portions of scapulae and femur from the same individual as YPM VPPU.021813 given the similar preservation of these elements (figure 6g).

*Ornithotarsus immanis* Cope 1869

**Material.** YPM 3221, a fragmentary hindlimb consisting of a distal tibia, distal fibula and fused astragalus and calcaneum.

**Locality and Horizon.** Southern shoreline of Raritan Bay near Keyport, Monmouth County, New Jersey; Late Santonian to Middle Campanian (approx. 84.3–77.8 Ma) [47] Merchantville Formation.

**Description.** The holotype of *O. immanis* includes the distal tibia (figure 8) and fibula, astragalus and fused calcaneum of a large hadrosaurid [42]. Prieto-Márquez *et al.* [49] briefly described this set of bones and considered *O. immanis* to be an indeterminate hadrosaurid, an assignment that is considered correct here. The distal tibia of *Ornithotarsus* is very similar to those of *Hadrosaurus foulkii* and other hadrosaurids (e.g. [49,99,101]). As in all hadrosaurids, the medial malleolus is large, triangular and strongly divergent, and does not extend as far distally as the lateral malleolus, the fibula is distally expanded, and the calcaneum is fused to the astragalus [49]. Unlike *H. foulkii* [49], *Gilmoreosaurus* [88] and *Probrachylophosaurus* [81], but as in *Bactrosaurus* [93], the lateral malleolus possesses a defined medial surface for articulation with the ascending process of the astragalus. The distal end of the lateral malleolus is also flattened in *Ornithotarsus*, whereas it is gently curved in *H. foulkii* [49]. Although these are minor differences, they do not favour the hypothesis proposed by Gallagher [42] and others that *O. immanis* is the same as *H. foulkii*. The apparent absence of diagnostic features on the holotype of *O. immanis* means that this name should only really be used if a more complete skeleton is recovered with a tibia, fibula and astragalus virtually identical to the holotype or which demonstrates the taxonomic utility of a previously overlooked feature in YPM 3321. I regard *O. immanis* as an indeterminate large hadrosaurid. Several other isolated bones from the Merchantville Formation, including a large caudal centrum YPM VPPU.22417 and a pedal phalanx YPM VPPU.022430, belong to hadrosauromorphs of similarly large size.

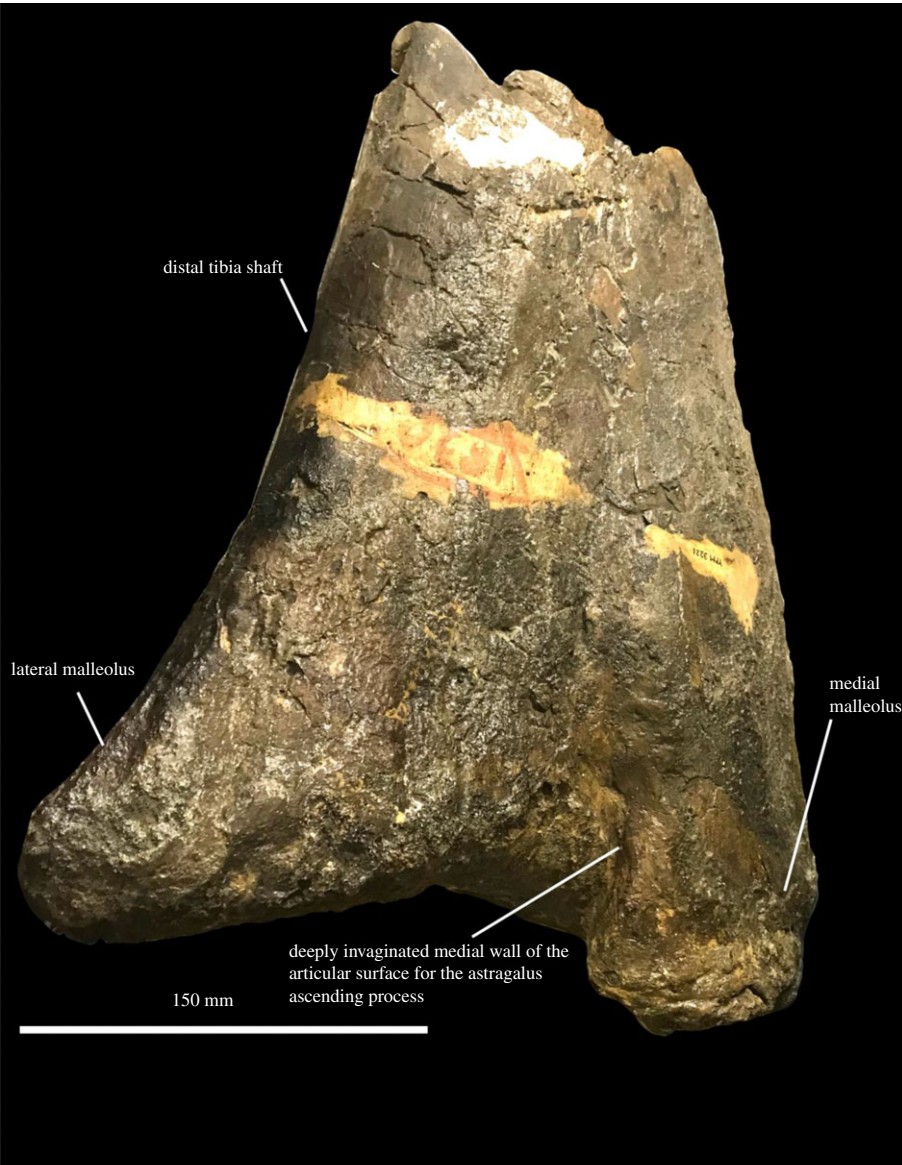

**Figure 8.** Anatomy of *Ornithotarsus immanis*. Tibia of YPM 3221 in (*a*) ventral view. Courtesy of the Division of Vertebrate Paleontology; Peabody Museum of Natural History, Yale University, New Haven, Connecticut, USA; peabody.yale.edu.

## 3.2. Phylogenetic results

The initial search of the TWiG dataset found 12 MPTs of 3312 steps. Branch swapping generated more than 100 000 MPTS. The strict consensus topology (TL = 3312; CI = 0.326, RI = 0.769) placed the Merchantville tyrannosauroid deep within Tyrannosauroidea in a polytomy with Eutyrannosauria and *Eotyrannus lengi*. The bootstrap tree features a support value of 13 for Eutyrannosauria, which may reflect the incompleteness of the Merchantville form. Characters uniting the Merchantville tyrannosauroid with derived tyrannosauroids include 200 (2: metatarsal III very pinched and hidden from view) and 728 (1: ratio between anteroposterior long axis and mediolateral width of the distal end of metatarsal IV is less than 1.4).

The initial search of the tyrannosauroid dataset found 40 MPTs, and branch-swapping found 164. The initial MPTs were of 799 steps each. The branch-swapped MPTs were also of length 799. The MPTs each had a consistency index of 0.564 and a retention index of 0.801. The Merchantville tyrannosauroid and *Dryptosaurus* form a monophyletic Dryptosauridae in a polytomy of a clade including *Appalachiosaurus* and tyrannosaurids and various derived Mid-Cretaceous tyrannosauroids. The clade Dryptosauridae is united by characters 387–390, which are newly added. However, the conditions that unite dryptosaurids are also found in other tyrannosauroids and theropods included

in the dataset. The consensus topology (figure 9*a*) was very similar to that found in previous analyses of this dataset (e.g. [10,12,28,31]), resolving Stokesosauridae and Proceratosauridae as the two basal clades of tyrannosaurs, a number of Early and Mid-Cretaceous tyrannosauroids as phylogenetically intermediate between these basal forms and the Eutyrannosauria [19], and a clade of large-bodied tyrannosaurs including the Appalachian forms and the Tyrannosauridae (e.g. [10,12,28,31]). The only major difference between the resolved consensus and previously published trees is the resolution of a dryptosaurid clade containing the Merchantville tyrannosauroid and *Dryptosaurus*, which is supported by the four characters listed above. The clade containing *Appalachiosaurus*, *Bistahieversor* and Tyrannosauridae is united by two characters: 372 (lateral malleolus of tibia strongly laterally divergent) and 382 (metatarsal II with a deep proximal excavation to articulate with metatarsal III; [28]). A metatarsal II with a deep proximal excavation to articulate with metatarsal III is also present in *Suskityrannus*, which is recovered as more basal than all other Mid-Cretaceous and Late Cretaceous tyrannosauroids [31]. This character is not coded for in *Moros*, the Iren Dabasu taxon, *Timurlengia*, or *Xiongguanlong*, which might explain why it does not appear as a synapomorphy for Eutyrannosauria in this analysis. Characters 379(1) and 387(1), which relate to metatarsal morphology, are recovered as synapomorphies for arctometatarsalian tyrannosauroids. Character 380, which relates to the appression of the metatarsals together, is recovered as a synapomorphy for all arctometatarsalian tyrannosauroids more derived than *Suskityrannus*. That a polytomy is formed at the base of Eutyrannosauria is likely reflective of the amount of missing data for three basal eutyrannosaurians (*Jinbeisaurus*, *Moros* and *Timurlengia*). For example, the removal of *Jinbeisaurus* resulted in a strict consensus tree wherein Dryptosauridae is sister to *Appalachiosaurus* + *Bistahieversor* + Tyrannosauridae.

The initial search of the hadrosauroid dataset found 39 MPTs, and resampling found 560 (electronic supplementary material, figure S5A–B). The initial MPTs were of 950 steps each. The bootstrap MPTs were also of length 950. The strict consensus (figure 9*b*) had a consistency index of 0.419 and a retention index of 0.771. Hadrosauridae is united by the following characters (descriptions after [23]): 18 (1: subrectangular/rectangular predentary oral denticles), 39 (1: rostrocaudally thin, strap-like rostral ascending process of the surangular wedges dorsally into thin sliver that is concealed in lateral view by the dorsal half of the caudal margin of the coronoid process), 48 (2: ventrally deflected and dorsoventrally expanded oral margin of premaxilla forms broadened lip), 49 (1: Premaxillary oral margin possesses external denticle-bearing layer and an internal layer of thickened bone set back from the oral margin and separated from the denticles by a deep, foramina-bearing sulcus), 168 (1: caudal margin of external naris solely formed by nasal), 182 (1: orbital dorsoventrally deeper than wide), 193 (1: oval supratemporal fenestra dorsal outline, wider mediolaterally than rostrocaudally), 199 (1: Angle between the lateral margins of glenoid and scapular articular surfaces on coronoid up to 115°), 202 (1: caudoventrally recurved ventral process of coronoid), 206 (1: broadened scapular neck greater than 60% of the dorsoventral depth of the scapular head), 226 (0: deep ilium central plate), 229 (3: craniocaudally shortened supraacetabular process of ilium), 242 (1: lateral profile of dorsal margin of ilium depressed over supraacetabular process and bowed over preacetabular process), 248 (1: lateral margin of iliac peduncle disappears ventrally into the lateral surface of the region adjacent to the acetabular margin of the pubis), 251 (1: presence of a lateroventral protuberance on the proximal region of the ischiadic peduncle of the pubis) and 258 (2: ischium pubic peduncle proximodistally shorter than the dorsoventral width of the distal articular surface). All commonly resolved groups in Hadrosauromorpha, including a primitive clade containing North American hadrosauromorphs like *Eolambia* and *Protohadros*, the Saurolophinae, and the Lambeosaurinae, are resolved as in previous studies [23]. The Merchantville hadrosaurid was found in a polytomy with *Hadrosaurus foulkii* and *Eotrachodon orientalis* at the base of Hadrosauridae. *Lophorhothon* and *Claosaurus* were placed outside of Hadrosauridae and form the three closest outgroups to that clade with *Tethyshadros insularis*.

In the analysis where the dentary and referred juvenile material was scored as a separate OTU from the partial skeleton, the partial skeleton nested with *Tethyshadros* as the sister clade to Hadrosauridae and the referred material was included in a large polytomy within Hadrosauridae.

# 4. Discussion

The Merchantville tyrannosauroid, hadrosaurid, and the indeterminate large hadrosaurid *Ornithotarsus immanis* constitute one of the only known late Santonian to early Campanian dinosaur assemblages

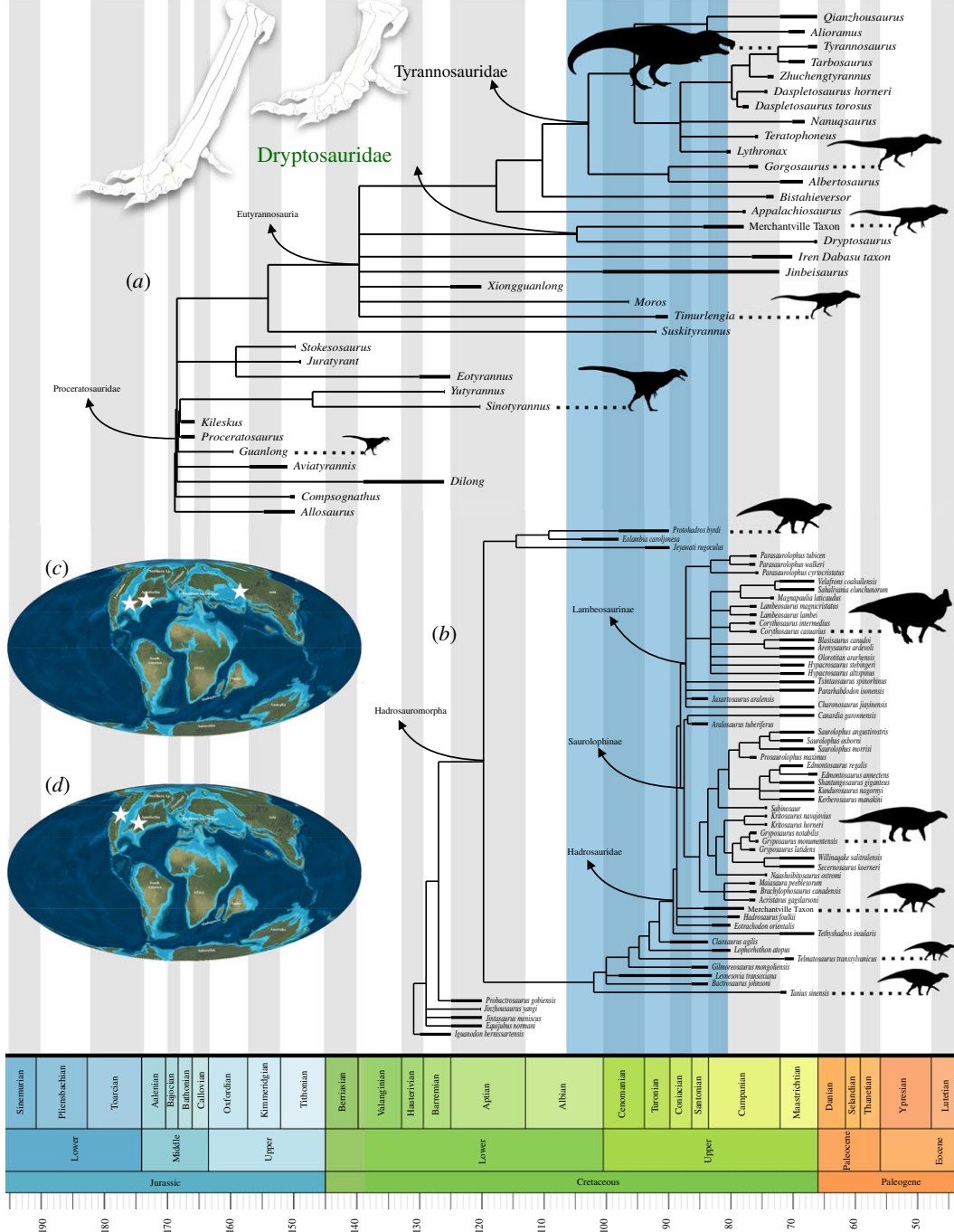

**Figure 9.** Spatio-temporal origins of latest Cretaceous dinosaur assemblages. (*a*) Time-calibrated phylogeny of tyrannosauroid dinosaurs showing the metatarsus conditions in Dryptosauridae and Tyrannosauridae. (*b*) Time-calibrated phylogeny of hadrosauromorph dinosaurs. Maps of Earth during the Early Campanian (approx. 80 Ma) (by R. Blakey, used with permission) showing the location of important Santonian eutyrannosaurian (*c*) and hadrosaurid (*d*) sites.

from North America. Among western North American units, only the Niobrara Formation of Kansas, the Milk River Formation of Alberta and the Allison Member (Menefee Formation) of New Mexico produce dinosaur material from the same time diagnostic to low taxonomic levels. Each has produced two species currently considered valid [102–107]. The Niobrara dinosaurs include the Santonian hadrosauromorph *Claosaurus* and nodosaurid *Niobrarasaurus* [102]. These are usually interpreted to come from Appalachia (e.g. [23]). The Allison Member assemblage includes material diagnostic to low taxonomic materials, and the two dinosaurs named from this unit are based on associated float [106,107]. The more substantial Milk River Formation assemblage includes exclusively Asian and western North

American faunal elements like pachycephalosaurs [104] in addition to a variety of indeterminate troodontid, tyrannosaurid and dromaeosaurid teeth [108].

Only a handful of dinosaur faunas from Eurasia and South America are thought to be of similar age to the Merchantville Formation. The Santonian–Campanian Javkhlant Formation of Mongolia and the apparently contemporaneous beds at Zos Canyon have produced the small ornithopod *Haya griva* [109,110], the ceratopsian *Yamaceratops dorngobiensis*, and undescribed theropod material [111]. Other Asian dinosaur assemblages, including those from the Iren Dabasu and Bayan Shireh Formations, include mid-sized tyrannosauroids like *Alectrosaurus* [72], large dromaeosaurids like *Achillobator* [112], and hadrosauromorphs like *Gilmoreosaurus* [88] and *Gobihadros* [92]. These formations may date to the Coniacian–Santonian (e.g. [113]) or alternatively much earlier (Cenomanian; [114]) or later (Campanian–Maastrichtian; [115,116]) in time, making comparisons with the Merchantville Formation tenuous. A handful of dinosaurs, including nodosaurids and ceratopsians, have been also described from the Santonian of Hungary [117]. Finally, the Bajo de la Carpa Formation in Argentina (e.g. [118–126]) preserves an excellent record of Santonian dinosaurs, but these include groups like abelisaurids and elasmarians which are entirely unknown in North America and Asia.

The Merchantville assemblage, which is contemporaneous with the lower Mooreville Chalk Formation in the southeastern USA [23,24], is therefore one of the only temporally constrained Santonian–Campanian dinosaur faunas and provides substantial new information about regional faunal differences. Whereas non-tyrannosaurid eutyrannosaurs like the Merchantville form and *Dryptosaurus* persisted in Appalachia, tyrannosaurines are known from the same time in western North America [16,106,108]. Although possible hadrosaurid material is known from the latest Santonian to early Campanian Milk River Formation, fossils recovered from this unit potentially show similarities with *Lophorhothon* [105], which is likely a non-hadrosaurid hadrosauromorph [23,79,92,94]. Whereas the presence of hadrosaurids in the western North American landmass Laramidia (figure 1*a*) by the Santonian is currently unclear, *Eotrachodon*, *Hadrosaurus*, and the Merchantville hadrosaurid show this clade was present and widespread by this time in Appalachia (figure 9*b*).

The description of a potentially novel hadrosaurid from the Santonian–Campanian of the Atlantic coastline also evidences the similarity of Late Cretaceous dinosaur faunas from Appalachia and Eurasia. Previous comparisons of Appalachian and Eurasian vertebrate assemblages highlighted the similarity of lambeosaurine, kogaionid, marsupial and crocodyliform fossils from these regions [117,127,128], but the low number of associated skeletons from Appalachia precluded testing these similarities under a phylogenetic framework. Putative Appalachian records of some of these clades, including the Lambeosaurinae, may also not show clear affinities to those groups [129]. Along with *Lophorhothon*, *Claosaurus*, *Hadrosaurus* and *Eotrachodon*, the Merchantville hadrosaurid forms a nearly continuous grade of derived hadrosauroids and early hadrosaurids restricted to Appalachia that is only interrupted by *Tethyshadros insularis* and *Telmatosaurus transsylvanicus* from the latest Cretaceous of Europe (fig. 10*b*; [23,89]). *Gobihadros mongoliensis* from the Early Late Cretaceous Bayan Shireh of Mongolia [92] and *Zhanghenglong yangchengensis* from the Santonian Majiacun Formation of China [130] were found in similar positions among derived hadrosauromorphs. The ubiquity of these derived non-hadrosaurid hadrosauromorphs in early Late Cretaceous assemblages suggests a degree of vicariance between Laramidian dinosaur faunas and others from the Northern Hemisphere that was probably prompted by the geographical isolation of western North America as the landmass Laramidia during the Late Cretaceous (figure 1*a*; e.g. [16,22–24,89]). Further discoveries will be needed to confirm the presence of clades of hadrosaurids endemic to Appalachia.

As the only tyrannosauroid known from the Santonian to earliest Campanian, the Merchantville form helps to fill a discontinuity in the tyrannosauroid record when tyrannosauroids were gaining in size [8,10,12,13]. Very recently, a record of early Middle Campanian tyrannosaurids has emerged from western North America, but these all are members of derived lineages with Tyrannosauridae [59,106,131]. By the Late Turonian, derived tyrannosaurs had not yet reached the dimensions of latest Cretaceous forms in western North America [10,13,132], suggesting tyrannosauroid evolution involved a very rapid period of body size increase. Several studies have suggested that tyrannosauroids filled niches left vacant by large-bodied allosaurs in the Middle Cretaceous based on the apparent increase in tyrannosauroid body size and the evolution of important features of the braincase and foot immediately following the extinction of Allosauroidea in the Northern Hemisphere [8,10,12,13,132]. The Merchantville form, which possessed metatarsals comparable in length to larger tyrannosauroids from the Late Cretaceous [16,54,59] further constrains this period by suggesting that large size was ancestral for dryptosaurids, tyrannosaurids, and their closest relatives and evolved by the Santonian.

This likely pushes the development of large body sizes in tyrannosaurs into the Coniacian and early Santonian, an interval of approximately 4 Myr.

Dryptosauridae is recovered in the same phylogenetic position as *Dryptosaurus* has been in over a decade's worth of phylogenetic analyses (figure 9*a*; e.g. [10,12,13,16,22,28,30,31,59,106,131]). The existence of a group of early diverging tyrannosauroids endemic to eastern North America has been hypothesized for over a century (e.g. [42,132–134]), but no strong evidence for a clade has been found until now. Although the position of the Appalachian tyrannosaur *Appalachiosaurus montgomeriensis* remains unchanged from previous phylogenies [16,22,28], it nests within the clade including tyrannosaurids and *Bistahieversor* to the exclusion of other taxa. This result shows representatives of both lineages of large-bodied tyrannosaur persisted in Appalachia (figure 9). The recognition of a multitaxic Dryptosauridae including the Merchantville form and *Dryptosaurus* also provides an excellent example of the influence of regional geographical isolation on the evolution of a predatory dinosaur clade. Few examples of insular theropods showing bizarre modifications to their skeletal anatomy comparable to those seen in modern carnivores from islands or insular landmasses have been documented [135]. The recognition of a clade of tyrannosauroids endemic to Appalachia provides the first example of a dinosaur group known solely from that landmass.

Despite reaching the dimensions of Campanian and Maastrichtian Laramidian and Eurasian tyrannosauroids, the Merchantville form possesses an extremely elongated pes more comparable to those of older tyrannosauroids like *Moros intrepidus* [12] and *Suskityrannus hazelae* [13]. The recognition of the distinctive metatarsal anatomy of dryptosaurids (figure 3) demonstrates that the acquisition of megapredatory roles by tyrannosauroids was not contingent on the development of features like a skull capable of osteophagy and a robust arctometatarsalian foot with a strongly reduced proximal end of metatarsal III [10,12,13]. The metatarsals of dryptosaurids neither closely resemble the arctometatarsalian foot of western North American and Asian tyrannosaurids (e.g. [55,56,66,67]), nor that of the Appalachian taxon *Appalachiosaurus* [22]. *Dryptosaurus* shows other features, including a lightly built skull, ziphodont dentition, and a massive raptorial hand, that along with a distinctive metatarsus substantiate a unique body plan among Tyrannosauroidea [16]. Nonetheless, dryptosaurids apparently filled large predatory niches in Appalachia just as tyrannosaurids did in Laramidia and Asia.

Gould & Lewontin [136] famously noted the existence of 'just so' stories in evolutionary biology, and noted the various hypotheses then put forward about the function of the diminutive forelimbs of tyrannosaurids as an example. In the same vein, the anatomy of the Merchantville species and *Dryptosaurus*, which may form a monophyletic lineage that lived at exactly the same time as tyrannosaurids, filled large predator niches in ecosystems also dominated by potential prey species like hadrosaurids, ornithomimosaurs and ankylosaurs, and succeeded allosaurs as large predators in assemblages along the Atlantic Coastal Plain [137,138], warrants a critical look at current hypotheses about the reasons tyrannosaurids evolved large sizes, advances senses and odd postcranial features like a small, two-fingered arm so late in the Mesozoic.

# 5. Conclusion

I describe dinosaurs from the Late Cretaceous Merchantville Formation of eastern North America that provide information on the evolution of vertebrate faunas in the Santonian and Early Campanian and the composition of Appalachian terrestrial faunas. These dinosaurs include a mid-sized tyrannosauroid that indicates the existence of a distinct clade of large theropods endemic to Appalachia characterized by many features of the lower hindlimb. The other, known from the cranial and appendicular remains of several individuals of different ontogenetic stages, provides another example of an early diverging hadrosaurid from the eastern half of North America.

Data accessibility. All fossils studied are in the Yale Peabody Museum, American Museum of Natural History and New Jersey State Museum. All data are included in the electronic supplementary material Text and tar.gz file attached to this submission.

The data are provided in electronic supplementary material [139].

Competing interests. I declare I have no competing interests.

Funding. I received no funding for this study.

Acknowledgements. I thank D. Brinkman, D. Parris, D. Ehret, R. Pellegrini, R. O. Johnson and C. Mehling for access to specimens in their care, and Jamie Henderson for the photographs in figure 2*e,f* (Courtesy of the Division of Vertebrate Paleontology; YPM VPPU.022416, Peabody Museum of Natural History, Yale University, New Haven, Connecticut, USA; peabody.yale.edu).

J. Napoli graciously provided photographs of the AMNH dentary. I thank R. Denton and J. Napoli for initial reviews of an earlier draft of this paper, and K. Davis, S. Brusatte, T. Carr and some anonymous reviewers for their comments on previous drafts. I acknowledge the Willi Hennig society for sponsoring the TNT phylogenetics program used for this study, and I thank R. and D. Blakey for allowing me to use the paleogeographic map in figure 6 (© Colorado Plateau Geosystems Inc.). Finally, I thank the communities at Pierrepont School and Yale College for their support and encouragement as I conducted the research presented in this paper.

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
