## [Peer Review File · Royal Society Open Science]

Review History

RSOS-210127.R0 (Original submission)

Review form: Reviewer 1

Is the manuscript scientifically sound in its present form?

Yes

Are the interpretations and conclusions justified by the results?

Yes

Is the language acceptable?

Yes

Do you have any ethical concerns with this paper?

No

Have you any concerns about statistical analyses in this paper?

Yes

Recommendation?

Accept with minor revision (please list in comments)

Comments to the Author(s)

I have now read the revised manuscript from Brownstein, as well as their reviewer responses.

While I remain personally unconvinced about the need to formally name this as a new tyrannosauroid taxon, the data the author has added certainly strengthens their argument, so I won't object to it further. I think some issues remain with the quantitative/morphometric analyses however (detailed below), so when proceeding with naming this taxon I think they should lean primarily on their comparative descriptions and qualitative results to frame their argument, doubly so given the revised descriptions and improved emphasis on hypothesized autapomorphies in the manuscript.

In terms of the specific changes in the article since the last version, I think the Discussion is much improved, particularly with respect to addressing issues raised with Appalachian assemblages being described as refugia, and in their reframing of the biogeography discussion.

Regarding the quantitative analyses, the issues I raised in the last review were largely unaddressed in the revised version. In the review response, the author acknowledges the issues in the analyses (particularly in this case the morphometrics and robusticity index comparisons), but their response of adding a few additional specimens does not actually address the issues that I described. To be clear, I don't think the issues in the quantitative analyses are make-or-break for this study, and I think these analyses could be minimized or removed, with the focus kept on the comparative description and biogeography. If the author wanted to keep the quantitative analyses though, they will need some work, which I will go through below.

1) Robusticity Index

The issues raised about the robusticity analysis relate to the data quality and variability. Data for each species derive from point samples of single individuals, with the exception of *Albertosaurus* (where $N=5$). The author argues that the lower robusticity index value associated with their new taxon is reflective of phylogenetic position, with it being most similar to older tyrannosauroids. However, the robusticity index results provided in Figure 3A show that *Cryptotyrannus* is about as similar in RI to basal tyrannosauroids as it is to later tyrannosaurids like *Albertosaurus*. As well, the 5 sampled individuals of *Albertosaurus* demonstrate that a substantial variation in RI can be expected in a single taxon (spanning about 40% of the total range of RI values recorded across all taxa in the analysis). While it is possible that this measure can be used to some effect in contrasting non-tyrannosauroids and tyrannosauroids (as argued in Peacock et al 2014), the large amount of variation and equivocal positioning of *Cryptotyrannus* in the overall RI distribution would suggest that this analysis adds little to no support to the author's arguments.

To illustrate my point, consider that in the text (lines 312-325) the authors argue that the metatarsus of *Cryptotyrannus* is most similar in robusticity to taxa like *Alectrosaurus* and distinct from the more robust condition shown in tyrannosaurids. Looking at the measured RI values, *Cryptotyrannus* has an RI value that is different from *Albertosaurus* by 0.027, and different from *Alectrosaurus* by 0.016. So the statement is technically true that *Cryptotyrannus* is more similar to *Alectrosaurus* than to most tyrannosaurids, but it is also misleading for several reasons. First, this difference represents not only a very small difference in the sense of the overall range of RI measured across the various species sampled, but also that within the only species where data for more than one individual exists (*Albertosaurus*), there is a range of RI of ~ 0.09 (from 0.32 to 0.41). The range of variation within this taxon alone is more than 3x the magnitude of difference that

exists separating *Cryptotyrannus* from *Albertosaurus*. As well, the closest actual RI value to *Cryptotyrannus* is not even *Alectrosaurus*, but rather a juvenile of *Bistahieversor* (RI of .295 vs .297)

Relatedly, the authors in the manuscript text also argue that *Cryptotyrannus* is very similar in robusticity to 'older tyrannosauroids' like *Moros* or *Suskityrannus* while referring to Figure 3A. As neither of those taxa are included in the RI analysis, I think this statement is at best a misleading reference to *Cryptotyrannus* being most similar in the RI analysis to 'older tyrannosauroids' like *Alectrosaurus*, though again as noted above it is actually most similar to a juvenile *Bistahieversor* (which although is a tyrannosauroid, isn't exactly 'older', occurring in younger rocks than *Cryptotyrannus* does), and fairly close in RI between *Alectrosaurus* and *Albertosaurus* specimens.

With all of that in mind, I think the robusticity analysis should probably be removed. I think at best it is unrelated or unresponsive of your argument that *Cryptotyrannus* is a distinct taxon, and at worst is actively working against your argument.

2) Linear Morphometrics & PCA

The issues raised with the linear morphometrics & PCA related primarily to the results not accounting for size-related effects. The authors may have added in some new measurements and specimens, but it hasn't fundamentally altered that problem. Looking at the PC data in the supplementary information, PC1 in the revised analysis now explains 87% of the variance (compared to 85% in the original analysis), with PC2 explaining about 7% of the variance (down from 12% in the original version). So even with the measurements from the three ornithomimosaurids & the juvenile tyrannosauroid, you're still getting results which are almost entirely reflecting the absolute size differences of the specimens (primarily in length, to a lesser extent circumference). This is also reflected in the loadings, where you can see that length and circumference are the most strongly related measured to PC1 and PC2. Consequently, the resulting plots don't provide a result that supports the argument that this is a distinct species based on various morphological characteristics, it just shows that it is most similar in size to a juvenile specimen of *Bistahieversor*. *Moros* remains an outlier on PC2 (though consistent with the overall size-driven pattern on PC1), but as about half of the measurements for *Moros*' are missing data (particularly for values like circumference and proximal transverse width which are strongly related to the PC2 loadings), it is difficult to fully characterize what is going on there. As an aside, the author doesn't explain in the Methods section what they did to account for missing measurements in their PCA. There are several ways to deal with missing data, but it's unclear here which of those they did. In any event though, the morphometric results still have the issue I noted last time, namely that as they are almost entirely related to the size of the specimen being measured, they aren't providing anything informative to support your argument of *Cryptotyrannus* being a distinct taxon. I suppose you could use these results to argue that it's something very similar to *Bistahieversor*, and/or the juvenile of a larger more robust tyrannosauroid, but I suspect you would prefer not to do so, given that it would work against the argument from the qualitative data / comparative description that this material represents a new taxon. To that end, I think you may want to consider removing the morphometric analyses. If you are going to keep them in, then I really need to emphasize the need to broaden them not by adding more taxa, but by plotting the results against a known size measurement you can account for in order to standardize things and hopefully tease out some more informative results about the shape differences between these sampled specimens.

Some more minor corrections:

Line 776: South America, not South American

Lines 824-828: I think there may be some words missing in this sentence

Lines 883-903: the font changes somewhat abruptly here, not sure if that is just an issue in my copy of the PDF, but perhaps best to confirm

Lines 893-903: perhaps could be interesting to compare and contrast what you're seeing between Laramidian tyrannosaurines (large size, big heads, small forelimbs) vs Appalachian dryptosaurids with the independent evolution of some of the features of the former group (large size, big heads, small forelimbs) in South American carcharodontosaurs while these animals similarly co-existed with distantly-related allosauroids that didn't go down that path. Not to imply that these groups were totally convergent, given that they did differ in some other features, such as growth rate (e.g. Cullen et al 2020), but there may be some similarity in the patterns underlying the evolution of those suites of characteristics, and how in both times/places you are seeing some large predator clades with those features and some related clades without them.

Review form: Reviewer 2 (Gregory Funston)

Is the manuscript scientifically sound in its present form?

No

Are the interpretations and conclusions justified by the results?

No

Is the language acceptable?

Yes

Do you have any ethical concerns with this paper?

No

Have you any concerns about statistical analyses in this paper?

Yes

Recommendation?

Reject

Comments to the Author(s)

While I find that the biogeographic issues have been solved by their removal, I still have issues with the taxonomy and the characters used to justify the identification of the theropod material. I remain unconvinced that the metatarsus can be referred to Tyrannosauroida, or that it is sufficiently diagnostic to erect a new taxon. The other reviewer raised these issues in the first submission and although my review of that version focussed on biogeography, I agree with their points.

My issues fall along two main lines, focusing on the theropod section. First, I believe broader comparisons are necessary to establish that the metatarsus is, in fact, tyrannosauroid. Again, I realize that this conclusion stems from earlier work, but seeing as this re-description overlaps significantly with the 2017 paper, I think re-examining these points is justified.

Second, I am unconvinced that the material is complete enough or the characters distinct enough from other tyrannosauroids to erect a new taxon. The author themselves commented on this issue in their 2017 paper, stating that "Based on the incomplete nature of YPM VPPU.021795, the author does not think it wise to name a new taxon based on this specimen." (Brownstein 2017, pg. 14). I do not see how the situation has changed. The only new specimen is the partial caudal

vertebra, which does not provide any characters to distinguish the material from other tyrannosaurs.

As a separate point, the proximal views of the metatarsals reveal labels “II” and “IV” that contradict the identification presented in the manuscript. Why were these identifications switched? I don’t see a comment on this in either this manuscript or the 2017 paper.

The author hinges the identification as a tyrannosaur on four main characters: 1) the presence of a deep, V-shaped notch on the proximal end of Metatarsal IV [also see previous comment]; 2) large, teardrop-shaped facets for Metatarsal III on the outer metatarsals; 3) an arctometatarsalian metatarsus where the proximal end of Metatarsal III is hidden in anterior view; and 4) visible muscle insertions on the posterior side of the metatarsals.

The author states that the four characters ally the specimen with eutyranosaur tyrannosauroids, and exclude it from other theropod groups. Each of these characters is problematic, however. With regards to the first character, a deep notch on metatarsal IV, there are several issues. This character seems to stem from Peacock et al. (2014), wherein they describe this notch in tyrannosaurs as ‘pointed’, not ‘deep’. First, this region of the specimen is heavily reconstructed and the depth of this notch may be overexaggerated. Second, if this is actually Metatarsal II, as the labelling in the figure shows, then the distribution of this character will have to be reassessed. Third, the author states that only two ornithomimid genera approach tyrannosaurs in robustness, but that they do not have a deep notch on Metatarsal IV. However, the author also shows that their specimen is less robust than other tyrannosaurs, so it is unclear why only the robust ornithomimids formed the comparison. Indeed, other ornithomimids, such as the species of *Ornithomimus*, do possess a deep notch on metatarsal IV (Claessens and Loewen 2015). Furthermore, nearly all arctometatarsalian theropods (caenagnathids, ornithomimids, troodontids, and tyrannosaurs) have a notch of some kind in this position, as noted by Peacock et al. (2014) in their initial description of the character. Without a quantitative analysis, it is unclear where the cutoff between ‘deep’ and ‘shallow’ might be, and it is also unclear from reading Peacock et al. (2014) whether the depth of the notch or its shape is the most critical factor. Considering the extensive reconstruction of this region, I do not think the shape can be confidently determined.

With regards to the second character, teardrop-shaped facets for Metatarsal III, I am not convinced that these can actually be identified in the specimen because of its incompleteness. Metatarsal II is broken distally, so the distal extent and shape of this facet cannot be determined. The same is partly true for Metatarsal IV, as well, and this region is extensively damaged, if not completely missing. Without being able to observe the entire extent of the buttressing surface, I am not sure how its shape can be confidently determined. I will agree that the shape does expand distally, but this is true of all arctometatarsalian theropods, to my knowledge. My interpretation of this character is that the key feature is the rounded distal terminus of the facet, as other theropods sometimes have pointed or oblique termini to the facet. However, I do not see how a rounded distal end of the facet could be established when these regions are so damaged. I even have trouble making out the proximal parts of the facet from the figures, and these could be better shown with an illustration or 3D model. Furthermore, the author specifically notes that caenagnathids differ from tyrannosaurs in having a “poorly developed” facet for Metatarsal III, but I disagree with this assessment, as those of caenagnathids are actually fairly prominent. The author claims that these facets ‘abut the distal condyles’, citing one of my figures as support for this assertion. However, that figure is not sufficiently detailed to allow this to be evaluated, and, indeed, this interpretation is false. The facets for Metatarsal III are separated from the distal condyles by a region of smooth bone as in all arctometatarsalian theropods. In that figure, this region exists where the shaft of the metatarsal is constricted in anterior view. I have also attached photographs of ROM 781, a caenagnathid metatarsus, that clearly show this (see Appendices A & B).

The third character, focused on the structure of the arctometatarsus, is perhaps the most problematic. The author states that tyrannosaurs are distinguished by an arctometatarsalian

structure in which Metatarsal III is hidden from the anterior surface at its proximal end. This has been considered a diagnostic feature of caenagnathids (Currie and Russell 1988, Currie 1989, Funston et al. 2016, Funston 2020), but in re-examining its distribution, I find it is also present in some ornithomimids (e.g. *Ornithomimus velox* Claessens et al. 2015), although less well-developed because metatarsal III is narrower at this point. Thus, this feature cannot be used to distinguish tyrannosaurs from caenagnathids or ornithomimids, and it may not even be useful for distinguishing any theropod family in its current phrasing.

Finally, *M. gastrocnemius lateralis* should be present in the same position in almost every theropod, although I am not sure whether its insertion varies in morphology between groups. However, it is difficult to assess the source for this character as all of the references for diagnostic characters are listed at the end of the sentence, rather than alongside the information they provided. Nonetheless, I am suspicious that differences in prominence of a scar of a muscle all theropods share might be more attributable to ontogeny or body mass, rather than taxonomy.

The author also provides some other features to distinguish the specimen from caenagnathids, but they have confused Metatarsals II and IV. The image they figure and the comparisons they draw are actually metatarsal IV, not metatarsal II. Metatarsal II can have a posterior process (e.g., Funston et al. 2015 Cret Res), although this is variable even within the group (and thus suggestive that it is too variable to clearly define differences between families). It is also worth noting that in their figure, the caenagnathid metatarsal (which is IV, not II) is covered by distal tarsal IV, which obscures the proximal shape.

As for the comparisons to other tyrannosaurs, some of the characters in the diagnosis are problematic in light of the contradictory labelling on the metatarsals. For example, the absence of a deep facet on the proximal surface of metatarsal II would be untrue if the labels shown in the figures are correct, as could the difference in proximal shape of metatarsal IV from *Dryptosaurus*. Finally, it is worth noting that many of the autapomorphies are simply the absence of the autapomorphies of *Dryptosaurus*, and these character states are widely distributed within Tyrannosauoidea (but not the two-taxon *Dryptosauridae*). Thus, most of these characters cannot be used to distinguish the taxon from other tyrannosaurs (e.g. the apex of the distal condyles, the mediolateral constriction, and the sulcus between the distal hemicondyles). In my opinion, these diagnostic characters are weak and may be oversplitting, by taking variation at face value with no consideration of allometric, ontogenetic or individual variation. This is especially problematic because of the fragmentary and reconstructed nature of the material. I agree with the other reviewer that the material is not sufficient to confidently state that the variation described is significant, without a broader analysis of individual and taxonomic variation within tyrannosaurs.

The PCA is insufficient, and needs to be size-scaled in some way. As mentioned by the other reviewer and even by the author in the description of the results, it is mostly size-driven. Ornithomimids clearly fall into the range of tyrannosaurs on the y-axis, so if size were eliminated, they may overlap considerably. I also do not think that including three ornithomimosaur points (but no other kinds of theropods) is sufficient, especially when there is significant variation within each group (e.g. *Gallimimus*, *Deinocheirus*, *Anserimimus*, which are all considerably different in metatarsal structure but aren't included). Furthermore, ornithomimosaur and other theropods are not included in the robusticity analysis, and the author has not responded to the other reviewer's criticisms of the use of robusticity index, which are quite valid. If anything, the quantitative analyses show to me that the author has not included enough comparisons in the study to be confident that the variation they show is meaningful.

To summarize my criticisms on the theropod part of the paper, I am not satisfied that the author has compared the material broadly enough to demonstrate that it pertains to any particular group of theropod dinosaurs. The characters they use are present in multiple groups but they restrict

the comparisons unnecessarily – particularly with regards to focusing on ornithomimids, and then only a subset of ornithomimids. Based on the description and figures, I am unconvinced that the material can be referred to a more specific clade than Coelurosauria indet., and I think erecting a new taxon for the material is unsupported.

Considering that I do not think the “tyrannosaur” taxon should be named, I do not see a logical connection between it and the rest of the manuscript. To me, it is mostly a rehash of the 2017 paper, and provides little beyond the inclusion into new phylogenies. Now that the biogeographic section has been removed, there is little to link the two sections of the manuscript. I think the tyrannosaur section should be dropped, and the focus should be placed solely on the hadrosaur material. To my eye, the hadrosaur section of the manuscript is much stronger because there is more material and more comparisons are drawn with other taxa.

Review form: Reviewer 3

Is the manuscript scientifically sound in its present form?

Yes

Are the interpretations and conclusions justified by the results?

Yes

Is the language acceptable?

Yes

Do you have any ethical concerns with this paper?

No

Have you any concerns about statistical analyses in this paper?

No

Recommendation?

Accept with minor revision (please list in comments)

Comments to the Author(s)

This is a welcomed and much needed contribution to our understanding of dinosaurian diversity, anatomy, relationships and biogeography in Appalachia, particularly given that the dinosaurian evolutionary history in Appalachia has been obscured by a patchy fossil record. The paper is generally well written and full of interesting and useful data. I cannot comment on the tyrannosauroid, as I am not an expert on this clade and I lack sufficient knowledge on tyrannosauroid anatomy to evaluate that part of the MS. I can, however, comment on the hadrosaurid. Also, I must say that I already reviewed an earlier version of this MS, previously submitted to another journal. Therefore, I do not have a lot to say this time, particularly because I see that my issues with that version have been addressed in the present one. I only have some minor remarks, listed below. Otherwise, I recommend publication of this paper pending those minor revisions.

Systematic Paleontology

Description

Line 530: "the presence of dental battery". Actually, non-hadrosaurid hadrosauroids also possess a dental battery. The difference is that this battery becomes more complex in hadrosaurids, having more than 30 tooth positions and with an occlusal plane reaching a maximum of three in width in the dentary dental battery.

Line 540-541: "The holotype and referred dentary likely come from individuals nearing or at osteological maturity....". You are not erecting a new genus or species, thus what holotype are you referring to? "Potentially" new hadrosaurid does not mean it is a new hadrosaurid.

Line 607: "The inside of the dentary is perforated by a hole.." Not hole, "foramen" is the word.

Line 737: "resampling". Do you mean Bootstrap?

Lines 739-741: It would be more convenient for the reader if you stated in words what characters are those.

Line 776: should be "South America", not "South American".

Line 810: not "evinces" but "evidences".

Figures

Figures 2, 4, 5, and 6 are missing. Maybe you forgot to upload them? Or a glitch of the manuscript site of the journal? Anyway, I cannot fully evaluate many of the characters of the osteology without seeing those images.

Figure S4B: The cladogram shows Atlantohadros, a genus name that does not exist as you are not erecting it (probably a remnant from an earlier version of the manuscript?).

Decision letter (RSOS-210127.R0)

Dear Mr Brownstein

The Editors assigned to your paper RSOS-210127 "Dinosaurs from a Santonian-Campanian coastal assemblage substantiate the enigmatic fauna of a North American paleolandmass" have now received comments from reviewers and would like you to revise the paper in accordance with the reviewer comments and any comments from the Editors. Please note this decision does not guarantee eventual acceptance.

We do not generally allow multiple rounds of revision so we urge you to make every effort to fully address all of the comments at this stage. If deemed necessary by the Editors, your

manuscript will be sent back to one or more of the original reviewers for assessment. If the original reviewers are not available, we may invite new reviewers.

Please submit your revised manuscript and required files (see below) no later than 21 days from today's (ie 04-May-2021) date. Note: the ScholarOne system will 'lock' if submission of the revision is attempted 21 or more days after the deadline. If you do not think you will be able to meet this deadline please contact the editorial office immediately.

on behalf of Dr Julia Brenda Desojo (Associate Editor) and Kevin Padian (Subject Editor)
openscience@royalsociety.org

Subject Editor Comments to Author (Professor Kevin Padian):

Comments to the Author:

Thanks for your resubmission. We have three somewhat different reviews here with some common threads and some strong concerns. First, the reviewers appreciate the improvement from the previous submission. And there are several remaining issues.

-- the two reviewers who feel confident to assess the theropod taxonomy don't agree that this has unique features that warrant naming a new species, and one reviewer is not convinced that it is even a tyrannosauroid.

-- one reviewer believes that it is better concentrate on the hadrosaurid material, and another offers some improvements in the interpretation of that material.

-- one reviewer is very concerned about the statistics and morphometrics and does not believe that they are appropriately applied and so they should be removed.

Our AE, without comment, has recommended "major revision." I will support that, and hope that you will reconsider some of the focus of your paper and whether it is really justified to name a new tyrannosaur. Whatever your decision, please notify our editorial office if you need more time to revise. Best wishes.

Reviewer comments to Author:

Reviewer: 1

Comments to the Author(s)

I have now read the revised manuscript from Brownstein, as well as their reviewer responses.

While I remain personally unconvinced about the need to formally name this as a new tyrannosauroid taxon, the data the author has added certainly strengthens their argument, so I won't object to it further. I think some issues remain with the quantitative/morphometric

analyses however (detailed below), so when proceeding with naming this taxon I think they should lean primarily on their comparative descriptions and qualitative results to frame their argument, doubly so given the revised descriptions and improved emphasis on hypothesized autapomorphies in the manuscript.

In terms of the specific changes in the article since the last version, I think the Discussion is much improved, particularly with respect to addressing issues raised with Appalachian assemblages being described as refugia, and in their reframing of the biogeography discussion.

Regarding the quantitative analyses, the issues I raised in the last review were largely unaddressed in the revised version. In the review response, the author acknowledges the issues in the analyses (particularly in this case the morphometrics and robusticity index comparisons), but their response of adding a few additional specimens does not actually address the issues that I described. To be clear, I don't think the issues in the quantitative analyses are make-or-break for this study, and I think these analyses could be minimized or removed, with the focus kept on the comparative description and biogeography. If the author wanted to keep the quantitative analyses though, they will need some work, which I will go through below.

1) Robusticity Index

The issues raised about the robusticity analysis relate to the data quality and variability. Data for each species derive from point samples of single individuals, with the exception of *Albertosaurus* (where $N=5$). The author argues that the lower robusticity index value associated with their new taxon is reflective of phylogenetic position, with it being most similar to older tyrannosauroids. However, the robusticity index results provided in Figure 3A show that *Cryptotyrannus* is about as similar in RI to basal tyrannosauroids as it is to later tyrannosaurids like *Albertosaurus*. As well, the 5 sampled individuals of *Albertosaurus* demonstrate that a substantial variation in RI can be expected in a single taxon (spanning about 40% of the total range of RI values recorded across all taxa in the analysis). While it is possible that this measure can be used to some effect in contrasting non-tyrannosauroids and tyrannosauroids (as argued in Peacock et al 2014), the large amount of variation and equivocal positioning of *Cryptotyrannus* in the overall RI distribution would suggest that this analysis adds little to no support to the author's arguments.

To illustrate my point, consider that in the text (lines 312-325) the authors argue that the metatarsus of *Cryptotyrannus* is most similar in robusticity to taxa like *Alectrosaurus* and distinct from the more robust condition shown in tyrannosaurids. Looking at the measured RI values, *Cryptotyrannus* has an RI value that is different from *Albertosaurus* by 0.027, and different from *Alectrosaurus* by 0.016. So the statement is technically true that *Cryptotyrannus* is more similar to *Alectrosaurus* than to most tyrannosaurids, but it is also misleading for several reasons. First, this difference represents not only a very small difference in the sense of the overall range of RI measured across the various species sampled, but also that within the only species where data for more than one individual exists (*Albertosaurus*), there is a range of RI of ~ 0.09 (from 0.32 to 0.41). The range of variation within this taxon alone is more than 3x the magnitude of difference that exists separating *Cryptotyrannus* from *Albertosaurus*. As well, the closest actual RI value to *Cryptotyrannus* is not even *Alectrosaurus*, but rather a juvenile of *Bistahieversor* (RI of .295 vs .297)

Relatedly, the authors in the manuscript text also argue that *Cryptotyrannus* is very similar in robusticity to 'older tyrannosauroids' like *Moros* or *Suskityrannus* while referring to Figure 3A. As neither of those taxa are included in the RI analysis, I think this statement is at best a misleading reference to *Cryptotyrannus* being most similar in the RI analysis to 'older tyrannosauroids' like *Alectrosaurus*, though again as noted above it is actually most similar to a juvenile *Bistahieversor* (which although is a tyrannosauroid, isn't exactly 'older', occurring in

younger rocks than *Cryptotyrannus* does), and fairly close in RI between *Alectrosaurus* and *Albertosaurus* specimens.

With all of that in mind, I think the robusticity analysis should probably be removed. I think at best it is unrelated or unresponsive of your argument that *Cryptotyrannus* is a distinct taxon, and at worst is actively working against your argument.

2) Linear Morphometrics & PCA

The issues raised with the linear morphometrics & PCA related primarily to the results not accounting for size-related effects. The authors may have added in some new measurements and specimens, but it hasn't fundamentally altered that problem. Looking at the PC data in the supplementary information, PC1 in the revised analysis now explains 87% of the variance (compared to 85% in the original analysis), with PC2 explaining about 7% of the variance (down from 12% in the original version). So even with the measurements from the three ornithomimosaurs & the juvenile tyrannosauroid, you're still getting results which are almost entirely reflecting the absolute size differences of the specimens (primarily in length, to a lesser extent circumference). This is also reflected in the loadings, where you can see that length and circumference are the most strongly related measured to PC1 and PC2. Consequently, the resulting plots don't provide a result that supports the argument that this is a distinct species based on various morphological characteristics, it just shows that it is most similar in size to a juvenile specimen of *Bistahieversor*. *Moros* remains an outlier on PC2 (though consistent with the overall size-driven pattern on PC1), but as about half of the measurements for *Moros*' are missing data (particularly for values like circumference and proximal transverse width which are strongly related to the PC2 loadings), it is difficult to fully characterize what is going on there. As an aside, the author doesn't explain in the Methods section what they did to account for missing measurements in their PCA. There are several ways to deal with missing data, but it's unclear here which of those they did. In any event though, the morphometric results still have the issue I noted last time, namely that as they are almost entirely related to the size of the specimen being measured, they aren't providing anything informative to support your argument of *Cryptotyrannus* being a distinct taxon. I suppose you could use these results to argue that it's something very similar to *Bistahieversor*, and/or the juvenile of a larger more robust tyrannosauroid, but I suspect you would prefer not to do so, given that it would work against the argument from the qualitative data / comparative description that this material represents a new taxon. To that end, I think you may want to consider removing the morphometric analyses. If you are going to keep them in, then I really need to emphasize the need to broaden them not by adding more taxa, but by plotting the results against a known size measurement you can account for in order to standardize things and hopefully tease out some more informative results about the shape differences between these sampled specimens.

Some more minor corrections:

Line 776: South America, not South American

Lines 824-828: I think there may be some words missing in this sentence

Lines 883-903: the font changes somewhat abruptly here, not sure if that is just an issue in my copy of the PDF, but perhaps best to confirm

Lines 893-903: perhaps could be interesting to compare and contrast what you're seeing between Laramidian tyrannosaurines (large size, big heads, small forelimbs) vs Appalachian dryptosaurids with the independent evolution of some of the features of the former group (large size, big heads, small forelimbs) in South American carcharodontosaurs while these animals similarly co-existed with distantly-related allosauroids that didn't go down that path. Not to imply that these groups were totally convergent, given that they did differ in some other features, such as growth rate (e.g. Cullen et al 2020), but there may be some similarity in the patterns

underlying the evolution of those suites of characteristics, and how in both times/places you are seeing some large predator clades with those features and some related clades without them.

Reviewer: 2

Comments to the Author(s)

While I find that the biogeographic issues have been solved by their removal, I still have issues with the taxonomy and the characters used to justify the identification of the theropod material. I remain unconvinced that the metatarsus can be referred to Tyrannosauroidea, or that it is sufficiently diagnostic to erect a new taxon. The other reviewer raised these issues in the first submission and although my review of that version focussed on biogeography, I agree with their points.

My issues fall along two main lines, focusing on the theropod section. First, I believe broader comparisons are necessary to establish that the metatarsus is, in fact, tyrannosauroid. Again, I realize that this conclusion stems from earlier work, but seeing as this re-description overlaps significantly with the 2017 paper, I think re-examining these points is justified. Second, I am unconvinced that the material is complete enough or the characters distinct enough from other tyrannosauroids to erect a new taxon. The author themselves commented on this issue in their 2017 paper, stating that "Based on the incomplete nature of YPM VPPU.021795, the author does not think it wise to name a new taxon based on this specimen." (Brownstein 2017, pg. 14). I do not see how the situation has changed. The only new specimen is the partial caudal vertebra, which does not provide any characters to distinguish the material from other tyrannosaurs.

As a separate point, the proximal views of the metatarsals reveal labels "II" and "IV" that contradict the identification presented in the manuscript. Why were these identifications switched? I don't see a comment on this in either this manuscript or the 2017 paper.

The author hinges the identification as a tyrannosaur on four main characters: 1) the presence of a deep, V-shaped notch on the proximal end of Metatarsal IV [also see previous comment]; 2) large, teardrop-shaped facets for Metatarsal III on the outer metatarsals; 3) an arctometatarsalian metatarsus where the proximal end of Metatarsal III is hidden in anterior view; and 4) visible muscle insertions on the posterior side of the metatarsals.

The author states that the four characters ally the specimen with eutyranosaur tyrannosauroids, and exclude it from other theropod groups. Each of these characters is problematic, however. With regards to the first character, a deep notch on metatarsal IV, there are several issues. This character seems to stem from Peacock et al. (2014), wherein they describe this notch in tyrannosaurs as 'pointed', not 'deep'. First, this region of the specimen is heavily reconstructed and the depth of this notch may be overexaggerated. Second, if this is actually Metatarsal II, as the labelling in the figure shows, then the distribution of this character will have to be reassessed. Third, the author states that only two ornithomimid genera approach tyrannosaurs in robustness, but that they do not have a deep notch on Metatarsal IV. However, the author also shows that their specimen is less robust than other tyrannosaurs, so it is unclear why only the robust ornithomimids formed the comparison. Indeed, other ornithomimids, such as the species of *Ornithomimus*, do possess a deep notch on metatarsal IV (Claessens and Loewen 2015). Furthermore, nearly all arctometatarsalian theropods (caenagnathids, ornithomimids, troodontids, and tyrannosaurs) have a notch of some kind in this position, as noted by Peacock et al. (2014) in their initial description of the character. Without a quantitative analysis, it is unclear where the cutoff between 'deep' and 'shallow' might be, and it is also unclear from reading Peacock et al. (2014) whether the depth of the notch or its shape is the most critical factor. Considering the extensive reconstruction of this region, I do not think the shape can be confidently determined.

With regards to the second character, teardrop-shaped facets for Metatarsal III, I am not convinced that these can actually be identified in the specimen because of its incompleteness. Metatarsal II is broken distally, so the distal extent and shape of this facet cannot be determined. The same is partly true for Metatarsal IV, as well, and this region is extensively damaged, if not completely missing. Without being able to observe the entire extent of the buttressing surface, I am not sure how its shape can be confidently determined. I will agree that the shape does expand distally, but this is true of all arctometatarsalian theropods, to my knowledge. My interpretation of this character is that the key feature is the rounded distal terminus of the facet, as other theropods sometimes have pointed or oblique termini to the facet. However, I do not see how a rounded distal end of the facet could be established when these regions are so damaged. I even have trouble making out the proximal parts of the facet from the figures, and these could be better shown with an illustration or 3D model. Furthermore, the author specifically notes that caenagnathids differ from tyrannosaurs in having a “poorly developed” facet for Metatarsal III, but I disagree with this assessment, as those of caenagnathids are actually fairly prominent. The author claims that these facets ‘abut the distal condyles’, citing one of my figures as support for this assertion. However, that figure is not sufficiently detailed to allow this to be evaluated, and, indeed, this interpretation is false. The facets for Metatarsal III are separated from the distal condyles by a region of smooth bone as in all arctometatarsalian theropods. In that figure, this region exists where the shaft of the metatarsal is constricted in anterior view. I have also attached photographs of ROM 781, a caenagnathid metatarsus, that clearly show this.

The third character, focused on the structure of the arctometatarsus, is perhaps the most problematic. The author states that tyrannosaurs are distinguished by an arctometatarsalian structure in which Metatarsal III is hidden from the anterior surface at its proximal end. This has been considered a diagnostic feature of caenagnathids (Currie and Russell 1988, Currie 1989, Funston et al. 2016, Funston 2020), but in re-examining its distribution, I find it is also present in some ornithomimids (e.g. *Ornithomimus velox* Claessens et al. 2015), although less well-developed because metatarsal III is narrower at this point. Thus, this feature cannot be used to distinguish tyrannosaurs from caenagnathids or ornithomimids, and it may not even be useful for distinguishing any theropod family in its current phrasing.

Finally, *M. gastrocnemius lateralis* should be present in the same position in almost every theropod, although I am not sure whether its insertion varies in morphology between groups. However, it is difficult to assess the source for this character as all of the references for diagnostic characters are listed at the end of the sentence, rather than alongside the information they provided. Nonetheless, I am suspicious that differences in prominence of a scar of a muscle all theropods share might be more attributable to ontogeny or body mass, rather than taxonomy.

The author also provides some other features to distinguish the specimen from caenagnathids, but they have confused Metatarsals II and IV. The image they figure and the comparisons they draw are actually metatarsal IV, not metatarsal II. Metatarsal II can have a posterior process (e.g., Funston et al. 2015 Cret Res), although this is variable even within the group (and thus suggestive that it is too variable to clearly define differences between families). It is also worth noting that in their figure, the caenagnathid metatarsal (which is IV, not II) is covered by distal tarsal IV, which obscures the proximal shape.

As for the comparisons to other tyrannosaurs, some of the characters in the diagnosis are problematic in light of the contradictory labelling on the metatarsals. For example, the absence of a deep facet on the proximal surface of metatarsal II would be untrue if the labels shown in the figures are correct, as could the difference in proximal shape of metatarsal IV from *Dryptosaurus*. Finally, it is worth noting that many of the autapomorphies are simply the absence of the autapomorphies of *Dryptosaurus*, and these character states are widely distributed within Tyrannosauroidae (but not the two-taxon *Dryptosauridae*). Thus, most of these characters cannot be used to distinguish the taxon from other tyrannosaurs (e.g. the apex of the distal condyles, the mediolateral constriction, and the sulcus between the distal hemicondyles). In my opinion, these

diagnostic characters are weak and may be oversplitting, by taking variation at face value with no consideration of allometric, ontogenetic or individual variation. This is especially problematic because of the fragmentary and reconstructed nature of the material. I agree with the other reviewer that the material is not sufficient to confidently state that the variation described is significant, without a broader analysis of individual and taxonomic variation within tyrannosaurs.

The PCA is insufficient, and needs to be size-scaled in some way. As mentioned by the other reviewer and even by the author in the description of the results, it is mostly size-driven. Ornithomimids clearly fall into the range of tyrannosaurs on the y-axis, so if size were eliminated, they may overlap considerably. I also do not think that including three ornithomimosaur points (but no other kinds of theropods) is sufficient, especially when there is significant variation within each group (e.g. Gallimimus, Deinocheirus, Anserimimus, which are all considerably different in metatarsal structure but aren't included). Furthermore, ornithomimosaur and other theropods are not included in the robusticity analysis, and the author has not responded to the other reviewer's criticisms of the use of robusticity index, which are quite valid. If anything, the quantitative analyses show to me that the author has not included enough comparisons in the study to be confident that the variation they show is meaningful.

To summarize my criticisms on the theropod part of the paper, I am not satisfied that the author has compared the material broadly enough to demonstrate that it pertains to any particular group of theropod dinosaurs. The characters they use are present in multiple groups but they restrict the comparisons unnecessarily – particularly with regards to focusing on ornithomimids, and then only a subset of ornithomimids. Based on the description and figures, I am unconvinced that the material can be referred to a more specific clade than Coelurosauria indet., and I think erecting a new taxon for the material is unsupported.

Considering that I do not think the “tyrannosaur” taxon should be named, I do not see a logical connection between it and the rest of the manuscript. To me, it is mostly a rehash of the 2017 paper, and provides little beyond the inclusion into new phylogenies. Now that the biogeographic section has been removed, there is little to link the two sections of the manuscript. I think the tyrannosaur section should be dropped, and the focus should be placed solely on the hadrosaur material. To my eye, the hadrosaur section of the manuscript is much stronger because there is more material and more comparisons are drawn with other taxa.

Reviewer: 3

Comments to the Author(s)

This is a welcomed and much needed contribution to our understanding of dinosaurian diversity, anatomy, relationships and biogeography in Appalachia, particularly given that the dinosaurian evolutionary history in Appalachia has been obscured by a patchy fossil record. The paper is generally well written and full of interesting and useful data. I cannot comment on the tyrannosauroid, as I am not an expert on this clade and I lack sufficient knowledge on tyrannosauroid anatomy to evaluate that part of the MS. I can, however, comment on the hadrosaurid. Also, I must say that I already reviewed an earlier version of this MS, previously submitted to another journal. Therefore, I do not have a lot to say this time, particularly because I see that my issues with that version have been addressed in the present one. I only have some minor remarks, listed below. Otherwise, I recommend publication of this paper pending those minor revisions.

Systematic Paleontology
Description

Line 530: “the presence of dental battery”. Actually, non-hadrosaurid hadrosauroids also possess a dental battery. The difference is that this battery becomes more complex in hadrosaurids, having more than 30 tooth positions and with an occlusal plane reaching a maximum of three in width in the dentary dental battery.

Line 540-541: “The holotype and referred dentary likely come from individuals nearing or at osteological maturity...”. You are not erecting a new genus or species, thus what holotype are you referring to? “Potentially” new hadrosaurid does not mean it is a new hadrosaurid.

Line 607: “The inside of the dentary is perforated by a hole..” Not hole, “foramen” is the word.

Line 737: “resampling”. Do you mean Bootstrap?

Lines 739-741: It would be more convenient for the reader if you stated in words what characters are those.

Line 776: should be “South America”, not “South American”.

Line 810: not “evinces” but “evidences”.

Figures

Figures 2, 4, 5, and 6 are missing. Maybe you forgot to upload them? Or a glitch of the manuscript site of the journal? Anyway, I cannot fully evaluate many of the characters of the osteology without seeing those images.

Figure S4B: The cladogram shows *Atlantohadros*, a genus name that does not exist as you are not erecting it (probably a remnant from an earlier version of the manuscript?).

===PREPARING YOUR MANUSCRIPT===

Your revised paper should include the changes requested by the referees and Editors of your manuscript. You should provide two versions of this manuscript and both versions must be provided in an editable format:
 one version identifying all the changes that have been made (for instance, in coloured highlight, in bold text, or tracked changes);
 a 'clean' version of the new manuscript that incorporates the changes made, but does not highlight them. This version will be used for typesetting if your manuscript is accepted.
 Please ensure that any equations included in the paper are editable text and not embedded images.

If you have been asked to revise the written English in your submission as a condition of publication, you must do so, and you are expected to provide evidence that you have received

language editing support. The journal would prefer that you use a professional language editing service and provide a certificate of editing, but a signed letter from a colleague who is a native speaker of English is acceptable. Note the journal has arranged a number of discounts for authors using professional language editing services (<https://royalsociety.org/journals/authors/benefits/language-editing/>).

===PREPARING YOUR REVISION IN SCHOLARONE===

<https://royalsociety.org/journals/authors/author-guidelines/#supplementary-material> to

include a suitable title and informative caption. An example of appropriate titling and captioning may be found at https://figshare.com/articles/Table_S2_from_Is_there_a_trade-off_between_peak_performance_and_performance_breadth_across_temperatures_for_aerobic_sc_ope_in_teleost_fishes_/3843624.

Author's Response to Decision Letter for (RSOS-210127.R0)

See Appendix C.

RSOS-210127.R1 (Revision)

Review form: Reviewer 1

Is the manuscript scientifically sound in its present form?

Yes

Are the interpretations and conclusions justified by the results?

Yes

Is the language acceptable?

Yes

Do you have any ethical concerns with this paper?

No

Have you any concerns about statistical analyses in this paper?

No

Recommendation?

Accept with minor revision (please list in comments)

Comments to the Author(s)

As I noted in my previous review, I remain unconvinced by the authors arguments for erecting a new taxon here. My view remains largely unchanged upon reading this subsequent revision. This species description is based on highly fragmentary material, and there is little accounting for intra-specific / ontogenetic variability given the paucity of the available data. But, as I also noted in my prior review, the authors have improved their arguments compared to their initial submission, emphasized the hypothesized autapomorphies, noted several other cases of tyrannosauroid taxa being named based on similarly fragmentary materials, and have added at least some note that this description is potentially contingent on finding less fragmentary material to re-assess in the future, so I am not going to insist they refrain from erecting a new taxon. Indeed, I think our disagreement on this point perhaps simply reflects a difference in philosophical approach to science and degree of caution exercised by this author vs others, as

evinced by their prior semi-frequent record of naming (or attempting to name) new species under similar conditions (i.e. highly fragmentary materials and a lack of examination of intra-specific variability and ontogeny). In any event, as noted previously, I've made my concerns here noted, and I think that so long as the authors are upfront about the nature of this material, then it's fine to allow them to publish their hypothesis and name this taxon. In the future, as more material is discovered, the issue can be re-visited and critically examined in a more robust way.

I see that the authors have also now removed the problematic quantitative sections. That is probably for the best, given that their analytical results were equivocal at best and working against their arguments at worst. I think that collecting more data and doing some kind of more thorough analyses would ideally have been better than removing them outright, but that would have likely required more data to exist than currently does for their new taxon (which as noted above and previously is sort of the crux of the issue in the paper generally). As it stands, and as I noted in my prior review, the analyses weren't really adding much in the first place so it seems fine to remove them and make it clear that the hypothesis is driven on the qualitative assessments and comparisons.

The updated phylogenetic analysis is welcome, as are the additional comparisons discussions, though the very low support values in their trees (noted by the author as likely relating to the incompleteness of the taxon) nicely underscore the concerns noted above and by the other reviewer about the relatively low data quality here and its potential implications for naming a new taxon vs. describing it as *Dryptosauridae* of uncertain association / with some features that are potentially distinct from *Dryptosaurus*

Concerning the hadrosaur, since it is no longer being named, I think some minor changes to the MS text might be warranted, since it is still frequently being referred to as a new taxon and discussed in reference to its holotype, etc. It doesn't really have a holotype since you aren't erecting it as a new taxon. I'd just discuss the material as being from this formation, and potentially new due to the following suite of features (a,b,c), with a note that you are not formally naming it due to the fragmentary nature (and other reasons x,y,z). As well, the hadrosaur is still labelled as 'Atlantohadros' in the Supplementary Information (e.g. Fig S5). Clarifying the text this way is mostly for consistency of the taxonomic hypotheses being presented, and shouldn't diminish the author's discussions of the importance of characterizing the biodiversity of this otherwise poorly characterized time and place.

minor note: that font change issue is still present on page 38, lines 855-868 of the pdf.

Review form: Reviewer 2 (Gregory Funston)

Is the manuscript scientifically sound in its present form?

Yes

Are the interpretations and conclusions justified by the results?

Yes

Is the language acceptable?

Yes

Do you have any ethical concerns with this paper?

No

Have you any concerns about statistical analyses in this paper?

No

Recommendation?

Accept with minor revision (please list in comments)

Comments to the Author(s)

The manuscript re-describes a fragmentary theropod metatarsus, erecting a new taxon for the material, and presents new hadrosaurid specimens. Together, these specimens provide information on the evolution of these clades in the Santonian–Campanian, as well as revealing more of the fauna of Appalachia.

My personal taxonomic philosophy, developed through my own work on very fragmentary theropods, is to be overly cautious and only refer specimens to certain clades or taxa when I can confidently eliminate other potential confounds, like ontogeny, individual variation, or taphonomic distortion. However, I understand that not all palaeontologists approach taxonomy in this way, and that arguments can be made for both sides. Thus, while I personally disagree that the material is sufficient to confidently refer to a tyrannosaur or to erect a new taxon, this is based on my philosophy, and I can provide no evidence against the author's hypothesis. I believe they have done as good a job as can be done to support their assertions with our current knowledge of these dinosaurs. Therefore, I believe the manuscript is fit to publish with some very minor revisions, outlined below:

Throughout: As this is being submitted to Royal Society Open Science, a UK journal, distances (miles) should be presented in metric units, or at least metric conversions should be presented in parentheses alongside Imperial units. The same is true of words with different spellings, e.g. palaeontology, colour, etc.

Throughout: In-text citations will need to be changed to RSOS's style.

Line 208: the abbreviation C&D Canal should be presented in parentheses after the first use of the full name. However, seeing as this abbreviation is only used twice, maybe it should just be spelled out in full?

Line 292–293: A word is missing at the end of line 292 "characteristic of Peacock et al., 2014;"

Line 486: "Monospecies" should be "Monospecific" or "Monodominant".

Decision letter (RSOS-210127.R1)

Dear Mr Brownstein

The Editors assigned to your paper RSOS-210127.R1 "Dinosaurs from a Santonian-Campanian coastal assemblage substantiate the enigmatic fauna of a North American paleolandmass" have now received comments from reviewers and would like you to revise the paper in accordance with the reviewer comments and any comments from the Editors. Please note this decision does not guarantee eventual acceptance.

Please submit your revised manuscript and required files (see below) no later than 21 days from today's (ie 23-Jun-2021) date. Note: the ScholarOne system will 'lock' if submission of the revision is attempted 21 or more days after the deadline. If you do not think you will be able to meet this deadline please contact the editorial office immediately.

on behalf of Prof Kevin Padian (Subject Editor)
openscience@royalsociety.org

Editor comments:

Thanks for your revisions. The reviewers are still not convinced of the validity of naming a new taxon, and I am not convinced that just because some people in the past have done it based on scrappy material that it is justifiable. However a new genus name will get into the literature and so is more likely to be searched than "Dryptosaurid indet.", and if the new name comes to be regarded as a nomen dubium it may ultimately find a home within another taxon. I'm submitting a "major revision" decision because the reviewers have identified some pervasive issues of format and consistency that need to be addressed fully and accurately, so as to save work for our editorial staff. Please make sure all of this is taken care of, or we will not be able to accept the submission. Good luck with your revisions.

Reviewer comments to Author:

Reviewer: 1

Comments to the Author(s)

As I noted in my previous review, I remain unconvinced by the authors arguments for erecting a new taxon here. My view remains largely unchanged upon reading this subsequent revision. This species description is based on highly fragmentary material, and there is little accounting for intra-specific / ontogenetic variability given the paucity of the available data. But, as I also noted in my prior review, the authors have improved their arguments compared to their initial

submission, emphasized the hypothesized autapomorphies, noted several other cases of tyrannosauroid taxa being named based on similarly fragmentary materials, and have added at least some note that this description is potentially contingent on finding less fragmentary material to re-assess in the future, so I am not going to insist they refrain from erecting a new taxon. Indeed, I think our disagreement on this point perhaps simply reflects a difference in philosophical approach to science and degree of caution exercised by this author vs others, as evinced by their prior semi-frequent record of naming (or attempting to name) new species under similar conditions (i.e. highly fragmentary materials and a lack of examination of intra-specific variability and ontogeny). In any event, as noted previously, I've made my concerns here noted, and I think that so long as the authors are upfront about the nature of this material, then it's fine to allow them to publish their hypothesis and name this taxon. In the future, as more material is discovered, the issue can be re-visited and critically examined in a more robust way.

I see that the authors have also now removed the problematic quantitative sections. That is probably for the best, given that their analytical results were equivocal at best and working against their arguments at worst. I think that collecting more data and doing some kind of more thorough analyses would ideally have been better than removing them outright, but that would have likely required more data to exist than currently does for their new taxon (which as noted above and previously is sort of the crux of the issue in the paper generally). As it stands, and as I noted in my prior review, the analyses weren't really adding much in the first place so it seems fine to remove them and make it clear that the hypothesis is driven on the qualitative assessments and comparisons.

The updated phylogenetic analysis is welcome, as are the additional comparisons discussions, though the very low support values in their trees (noted by the author as likely relating to the incompleteness of the taxon) nicely underscore the concerns noted above and by the other reviewer about the relatively low data quality here and its potential implications for naming a new taxon vs. describing it as *Dryptosauridae* of uncertain association / with some features that are potentially distinct from *Dryptosaurus*

Concerning the hadrosaur, since it is no longer being named, I think some minor changes to the MS text might be warranted, since it is still frequently being referred to as a new taxon and discussed in reference to its holotype, etc. It doesn't really have a holotype since you aren't erecting it as a new taxon. I'd just discuss the material as being from this formation, and potentially new due to the following suite of features (a,b,c), with a note that you are not formally naming it due to the fragmentary nature (and other reasons x,y,z). As well, the hadrosaur is still labelled as 'Atlantohadros' in the Supplementary Information (e.g. Fig S5). Clarifying the text this way is mostly for consistency of the taxonomic hypotheses being presented, and shouldn't diminish the author's discussions of the importance of characterizing the biodiversity of this otherwise poorly characterized time and place.

minor note: that font change issue is still present on page 38, lines 855-868 of the pdf.

Reviewer: 2

Comments to the Author(s)

The manuscript re-describes a fragmentary theropod metatarsus, erecting a new taxon for the material, and presents new hadrosaurid specimens. Together, these specimens provide information on the evolution of these clades in the Santonian-Campanian, as well as revealing more of the fauna of Appalachia.

My personal taxonomic philosophy, developed through my own work on very fragmentary theropods, is to be overly cautious and only refer specimens to certain clades or taxa when I can confidently eliminate other potential confounds, like ontogeny, individual variation, or

taphonomic distortion. However, I understand that not all palaeontologists approach taxonomy in this way, and that arguments can be made for both sides. Thus, while I personally disagree that the material is sufficient to confidently refer to a tyrannosaur or to erect a new taxon, this is based on my philosophy, and I can provide no evidence against the author's hypothesis. I believe they have done as good a job as can be done to support their assertions with our current knowledge of these dinosaurs. Therefore, I believe the manuscript is fit to publish with some very minor revisions, outlined below:

Throughout: As this is being submitted to Royal Society Open Science, a UK journal, distances (miles) should be presented in metric units, or at least metric conversions should be presented in parentheses alongside Imperial units. The same is true of words with different spellings, e.g. palaeontology, colour, etc.

Throughout: In-text citations will need to be changed to RSOS's style.

Line 208: the abbreviation C&D Canal should be presented in parentheses after the first use of the full name. However, seeing as this abbreviation is only used twice, maybe it should just be spelled out in full?

Line 292–293: A word is missing at the end of line 292 “characteristic of Peacock et al., 2014;”

Line 486: “Monospecies” should be “Monospecific” or “Monodominant”.

===PREPARING YOUR MANUSCRIPT===

===PREPARING YOUR REVISION IN SCHOLARONE===

Author's Response to Decision Letter for (RSOS-210127.R1)

See Appendix D.

RSOS-210127.R2 (Revision)

Review form: Reviewer 1

Is the manuscript scientifically sound in its present form?

Yes

Are the interpretations and conclusions justified by the results?

Yes

Is the language acceptable?

Yes

Do you have any ethical concerns with this paper?

No

Have you any concerns about statistical analyses in this paper?

No

Recommendation?

Accept as is

Comments to the Author(s)

I've reviewed the latest round of changes made by the author, and think the manuscript is suitable for acceptance/publication.

Review form: Reviewer 2 (Gregory Funston)

Is the manuscript scientifically sound in its present form?

Yes

Are the interpretations and conclusions justified by the results?

Yes

Is the language acceptable?

Yes

Do you have any ethical concerns with this paper?

No

Have you any concerns about statistical analyses in this paper?

No

Recommendation?

Accept as is

Comments to the Author(s)

I am pleased to see that the author has taken onboard all of the many changes we reviewers have suggested. I appreciate the author's patience and professionalism going through four rounds of revision on the manuscript, which I understand can be quite frustrating. The manuscript is now very sound and will be an important contribution to the field and is sure to be foundational to all future work on Appalachian dinosaurs. I have no further suggestions for the manuscript.

Decision letter (RSOS-210127.R2)

Dear Mr Brownstein,

It is a pleasure to accept your manuscript entitled "Dinosaurs from the Santonian-Campanian Atlantic coastline substantiate phylogenetic signatures of vicariance in Cretaceous North America" in its current form for publication in Royal Society Open Science. The comments of the reviewer(s) who reviewed your manuscript are included at the foot of this letter.

Kind regards,

Royal Society Open Science Editorial Office
Royal Society Open Science

on behalf of Professor Kevin Padian (Subject Editor)
openscience@royalsociety.org

Reviewer comments to Author:

Reviewer: 1

Comments to the Author(s)

I've reviewed the latest round of changes made by the author, and think the manuscript is suitable for acceptance/publication.

Reviewer: 2

Comments to the Author(s)

I am pleased to see that the author has taken onboard all of the many changes we reviewers have suggested. I appreciate the author's patience and professionalism going through four rounds of revision on the manuscript, which I understand can be quite frustrating. The manuscript is now very sound and will be an important contribution to the field and is sure to be foundational to all future work on Appalachian dinosaurs. I have no further suggestions for the manuscript.

Appendix A

14.01.2015 10:43

Appendix B

Appendix C

Cover Letter

May 7th, 2021

Dear *Royal Society Open Science* Editorial Board,

I would like to resubmit my paper “Dinosaurs from a Santonian-Campanian coastal assemblage substantiate the enigmatic fauna of a North American paleolandmass” for your consideration. I have revised the paper in line with the reviewers’ comments and hope the manuscript is now suitable for publication at your journal.

Particularly, I have made all the smaller corrections given by reviewers 1 and 3 and removed the morphometric analyses in order to fully address the points made by reviewer 1 and reviewer 2. I have also responded feature-by-feature to reviewer 2’s comments regarding the character list used to ally *Cryptotyrannus* with Tyrannosauroidae. I have also revised these lists in the manuscript and noted additional contrasts where applicable. Finally, in order to thoroughly test the affinities of *Cryptotyrannus*, I included it in the recently published large coelurosaurian matrix of Pei et al. (2020), which represents an extension of the 2014 TAFT matrix I originally used to test the affinities of *Cryptotyrannus* in 2017. This placed *Cryptotyrannus* in Eutyranosauria along with other arctometatarsalian Appalachian taxa.

Thank you for your consideration, and I look forward to your response.

Regards,

Chase Doran Brownstein Research Associate, Stamford Museum and Nature Center Undergraduate,
Department of Ecology and Evolutionary Biology, Yale University

Reviewer 1.

While I remain personally unconvinced about the need to formally name this as a new tyrannosauroid taxon, the data the author has added certainly strengthens their argument, so I won't object to it further. I think some issues remain with the quantitative/morphometric analyses however (detailed below), so when proceeding with naming this taxon I think they should lean primarily on their comparative descriptions and qualitative results to frame their argument, doubly so given the revised descriptions and improved emphasis on hypothesized autapomorphies in the manuscript.

Regarding the quantitative analyses, the issues I raised in the last review were largely unaddressed in the revised version. In the review response, the author acknowledges the issues in the analyses (particularly in this case the morphometrics and robusticity index comparisons), but their response of adding a few additional specimens does not actually address the issues that I described. To be clear, I don't think the issues in the quantitative analyses are make-or-break for this study, and I think these analyses could be minimized or removed, with the focus kept on the comparative description and biogeography. If the author wanted to keep the quantitative analyses though, they will need some work, which I will go through below.

Thanks for this comment. I agree with the reviewer's point. Given the comments made by the other reviewer, I have decided to remove these analyses in favor of providing additional comparative figures and description in the main text. Thus, the reviewer's recommendations of removing the robusticity and principal components analyses is accepted.

Some more minor corrections:

Line 776: South America, not South American

This has been corrected.

Lines 824-828: I think there may be some words missing in this sentence

Lines 883-903: the font changes somewhat abruptly here, not sure if that is just an issue in my copy of the PDF, but perhaps best to confirm

This seems to be an issue with the PDF.

Lines 893-903: perhaps could be interesting to compare and contrast what you're seeing between Laramidian tyrannosaurines (large size, big heads, small forelimbs) vs Appalachian dryptosauroids with the independent evolution of some of the features of the former group (large size, big heads, small forelimbs) in South American carcharodontosaurs while these animals similarly co-existed with distantly-related allosauroids that didn't go down that path. Not to imply that these groups were totally convergent, given that they did differ in some other features, such as growth rate (e.g. Cullen et al 2020), but there may be some similarity in the patterns underlying the evolution of those suites of characteristics, and how in both times/places you are seeing some large predator clades with those features and some related clades without them.

Reviewer: 2

I am unconvinced that the material is complete enough or the characters distinct enough from other tyrannosauroids to erect a new taxon. The author themselves commented on this issue in their 2017 paper, stating that "Based on the incomplete nature of YPM VPPU.021795, the author does not think it wise to name a new taxon based on this specimen." (Brownstein 2017, pg. 14). I do not see how the situation has changed. The only new specimen is the partial caudal vertebra, which does not provide any characters to distinguish the material from other tyrannosaurs.

As a separate point, the proximal views of the metatarsals reveal labels "II" and "IV" that contradict the identification presented in the manuscript. Why were these identifications switched? I don't see a comment on this in either this manuscript or the 2017 paper.

I appreciate these comments but have chosen to go along with the description of a new taxon based on the specimen. First, I note here and in previous responses to reviewers that the pes is often used to name new species of theropod dinosaur and is considered particularly diagnostic with respect to tyrannosauroids. The diagnoses of several tyrannosaurs, including *Suskityrannus*, *Moros*, *Alectrosaurus*, and *Dryptosaurus* all in (large) part rely on features of the metatarsus. So, I would push back against the idea that the specimen is too incomplete. Regarding the number of diagnostic features, my note would be that I actually recognize a similar number of diagnostic features on the holotype of *Cryptotyrannus* to counts currently recognized for *Timurlengia*, *Suskityrannus*, and *Moros*. The reason that I have chosen to continue with a diagnosis despite voicing uncertainty in 2017 has been (1) the publication of additional tyrannosaur pedal specimens that evidence the diagnostic nature of the region, (2) further examination of comparative materials, and (3) the denser sampling of tyrannosauroids in phylogenetic matrices.

Regarding the reviewer's concern about labeling, the "II" and "IV" on the metatarsals were added just after preparation, likely in the 1970s-80s. Although I do not know exactly why these were identified as such, these ID's are clearly incorrect. The metatarsal IV I identify has the clear, deep articular sulcus for a proximally crescentic metatarsal III, for example. Mt II is also slightly more robust.

The author hinges the identification as a tyrannosaur on four main characters: 1) the presence of a deep, V-shaped notch on the proximal end of Metatarsal IV [also see previous comment]; 2) large, teardrop-shaped facets for Metatarsal III on the outer metatarsals; 3) an arctometatarsalian metatarsus where the proximal end of Metatarsal III is hidden in anterior view; and 4) visible muscle insertions on the posterior side of the metatarsals.

The author states that the four characters ally the specimen with eutyranosaur tyrannosauroids, and exclude it from other theropod groups. Each of these characters is problematic, however. With regards to the first character, a deep notch on metatarsal IV, there are several issues. This character seems to stem from Peacock et al. (2014), wherein they describe this notch in tyrannosaurs as 'pointed', not 'deep'. First, this region of the specimen is heavily reconstructed and the depth of this notch may be overexaggerated. Second, if this is actually Metatarsal II, as the labelling in the figure shows, then the distribution of this character will have to be reassessed. Third, the author states that only two ornithomimid genera approach tyrannosaurs in robustness, but that they do not have a deep notch on Metatarsal IV. However, the author also shows that their specimen is less robust than other tyrannosaurs, so it is unclear why only the robust ornithomimids formed the comparison. Indeed, other ornithomimids, such as the species of *Ornithomimus*, do possess a deep notch on metatarsal IV (Claessens and Loewen 2015). Furthermore, nearly all arctometatarsalian theropods (caenagnathids, ornithomimids, troodontids, and tyrannosaurs) have a notch of some kind in this position, as noted by Peacock et al. (2014) in their initial description of the character. Without a quantitative analysis, it is unclear where the cutoff between 'deep' and 'shallow' might be, and it is also unclear from reading Peacock et al. (2014) whether the depth of the notch or its shape is the most critical factor. Considering the extensive reconstruction of this region, I do not think the shape can be confidently determined.

I appreciate this comment. However, I note that the presence of a deepened articular sulcus on metatarsal IV to accommodate a proximally crescentic metatarsal III has been recognized as a synapomorphy of Tyrannosauridae/more recently Eutyranosauria by every major analysis since at least the 1990s. This feature is a character in the major Brusatte/Carr/TWiG matrices for tyrannosauroids and has also been consistently employed to differentiate derived tyrannosauroid feet from other clades in the literature (e.g., Carr et al., 2005; Brusatte et al., 2011; Thompson et al., 2013; Peacock et al., 2014; Chinzorig et al., 2017; Nesbitt et al., 2019; Zanno et al., 2019). See above for the labeling issue on the specimens. As I have noted in the previous review of this paper, the reconstruction of the proximal end of the specimen consists of stabilization of the present bone. Bone still remains throughout the deep articular surface. This is also validated by other studies of the Merchantville metatarsus (e.g., Dalman et al., 2017 in *Memoirs of the Fukui Prefectural Museum*).

I also appreciate the comment regarding quantification of this feature. However, given the extensive literature supporting the distinction between the tyrannosauroid condition and that in other theropods, I still believe qualitative justifications are sufficient for the referral. The author notes that some robust-pes ornithomimids (i.e., *Ornithomimus*) have deepened metatarsal IV facets for metatarsal III. To illustrate the clear distinction, I've added a figure from Claessens and Loewen that includes traces of the articular facet for mt. III on mt IV:

gens and
en 2015

As shown above, the tyrannosauroid condition is clearly much deeper than that present in *Ornithomimus*, which arguably shows the deepest mt III facet for mt IV among arctometatarsalian ornithomimosaur.

With regards to the second character, teardrop-shaped facets for Metatarsal III, I am not convinced that these can actually be identified in the specimen because of its incompleteness. Metatarsal II is broken distally, so the distal extent and shape of this facet cannot be determined. The same is partly true for Metatarsal IV, as well, and this region is extensively damaged, if not completely missing. Without being able to observe the entire extent of the buttressing surface, I am not sure how its shape can be confidently determined. I will agree that the shape does expand distally, but this is true of all arctometatarsalian theropods, to my knowledge. My interpretation of this character is that the key feature is the rounded distal terminus of the facet, as other theropods sometimes have pointed or oblique termini to the facet. However, I do not see how a rounded distal end of the facet could be established when these regions are so damaged. I even have trouble making out the proximal parts of the facet from the figures, and these could be better shown with an illustration or 3D model. Furthermore, the author specifically notes that caenagnathids differ from tyrannosaurs in having a "poorly developed" facet for Metatarsal III, but I disagree with this assessment, as those of caenagnathids are actually fairly prominent. The author claims that these facets 'abut the distal condyles', citing one of my figures as support for this assertion. However, that figure is not sufficiently detailed to allow this to be evaluated, and, indeed, this interpretation is false. The facets for Metatarsal III are separated from the distal condyles by a region of smooth bone as in all arctometatarsalian theropods. In that figure, this region exists where the shaft of the metatarsal is constricted in anterior view. I have also attached photographs of ROM 781, a caenagnathid metatarsus, that clearly show this.

I believe this issue mainly just has to do with the angle of light in the photos taken for this contribution. See Brownstein (2017) figure 2 and dotted line below:

These are commonly considered synapomorphies of Tyrannosauroidea (see Holtz, 2004; Carr et al., 2005; Thompson et al., 2013; Peacock et al., 2014; Zanno et al., 2019). I have added an additional figure visualizing this in the main manuscript.

The third character, focused on the structure of the arctometatarsus, is perhaps the most problematic. The author states that tyrannosaurs are distinguished by an arctometatarsalian structure in which Metatarsal III is hidden from the anterior surface at its proximal end. This has been considered a

diagnostic feature of caenagnathids (Currie and Russell 1988, Currie 1989, Funston et al. 2016, Funston 2020), but in re-examining its distribution, I find it is also present in some ornithomimids (e.g. *Ornithomimus velox* Claessens et al. 2015), although less well-developed because metatarsal III is narrower at this point. Thus, this feature cannot be used to distinguish tyrannosaurs from caenagnathids or ornithomimids, and it may not even be useful for distinguishing any theropod family in its current phrasing.

I appreciate this comment. To respond to the reviewer, the closely appressed nature of the metatarsals and hidden proximal mt III just define the arctometatarsalian condition. This was not meant as a synapomorphy of Eutyranosauria among Coelurosauria but as an apomorphy of Eutyranosauria among Tyrannosauroida. This has been rephrased in the manuscript.

Finally, M. gastrocnemius lateralis should be present in the same position in almost every theropod, although I am not sure whether its insertion varies in morphology between groups. However, it is difficult to assess the source for this character as all of the references for diagnostic characters are listed at the end of the sentence, rather than alongside the information they provided. Nonetheless, I am suspicious that differences in prominence of a scar of a muscle all theropods share might be more attributable to ontogeny or body mass, rather than taxonomy.

I appreciate this comment. This has been rephrased to reflect the diagnostic eutyranosaurian condition, which is the presence of a concavity or ridge separating this large muscle attachment from the mt. III articulation. The size and development of the M. gastrocnemius lateralis attachment site is what differentiates this bone from the corresponding ones in Ornithomimids (e.g., Zanno et al., 2019). So, the character is the presence/absence of a ridge rather than the presence/absence of the rugose attachment site.

The author also provides some other features to distinguish the specimen from caenagnathids, but they have confused Metatarsals II and IV. The image they figure and the comparisons they draw are actually metatarsal IV, not metatarsal II. Metatarsal II can have a posterior process (e.g., Funston et al. 2015 Cret Res), although this is variable even within the group (and thus suggestive that it is too variable to clearly define differences between families). It is also worth noting that in their figure, the caenagnathid metatarsal (which is IV, not II) is covered by distal tarsal IV, which obscures the proximal shape.

I appreciate this comment and have revised the manuscript figure accordingly. I have also added text distinguishing the holotype of *Cryptotyrannus* from caenagnathids following Peacock et al. (2014): the presence of a slight, 'kinked' lateral divergence of the distal end of metatarsal IV.

As for the comparisons to other tyrannosaurs, some of the characters in the diagnosis are problematic in light of the contradictory labelling on the metatarsals.

See above.

*Finally, it is worth noting that many of the autapomorphies are simply the absence of the autapomorphies of *Dryptosaurus*, and these character states are widely distributed within Tyrannosauroida (but not the two-taxon *Dryptosauridae*). Thus, most of these characters cannot be used to distinguish the taxon from other tyrannosaurs (e.g. the apex of the distal condyles, the mediolateral constriction, and the sulcus between the distal hemicondyles). In my opinion, these diagnostic characters are weak and may be oversplitting, by taking variation at face value with no consideration of allometric, ontogenetic or individual variation. This is especially problematic because of the fragmentary and reconstructed nature of the material. I agree with the other reviewer that the material is not sufficient to confidently state that the variation described is significant, without a broader analysis of individual and taxonomic variation within tyrannosaurs.*

I appreciate this comment. However, I note that the apex of the distal condyles is herein recognized as a new autapomorphy of *Dryptosaurus*, and not *Cryptotyrannus*. As shown in figure S2, this region is flat in *Cryptotyrannus* and other tyrannosauroids. I also note that the shape of the metatarsal in proximal view is an autapomorphy of *Dryptosaurus* (Brusatte et al., 2011). Finally, the mediolateral compression of metatarsal IV in distal view was previously considered an autapomorphy of *Moros* (Zanno et al., 2019). This shows that another research team independently came to the conclusion that this character was genuinely diagnostic. I have removed the character involving the sulcus between the hemicondyles following this comment.

In order to provide further support that the characters observable for *Cryptotyrannus* unite it with eutyranosaurian tyrannosauroids over other coelurosaurs, I analyzed revised codings of it in the matrix of Pei et al. (2020), which includes a very large sample of all the groups we are discussing: Tyrannosauroida, Ornithomimosauria, Oviraptorosauria, Troodontidae, etc.

The PCA is insufficient, and needs to be size-scaled in some way. As mentioned by the other reviewer and even by the author in the description of the results, it is mostly size-driven. Ornithomimids clearly fall into the range of tyrannosaurs on the y-axis, so if size were eliminated, they may overlap considerably. I also do not think that including three ornithomimosaur points (but no other kinds of theropods) is sufficient, especially when there is significant variation within each group (e.g. Gallimimus, Deinocheirus, Anserimimus, which are all considerably different in metatarsal structure but aren't included). Furthermore, ornithomimosaur and other theropods are not included in the robusticity analysis, and the author has not responded to the other reviewer's criticisms of the use of robusticity index, which are quite valid. If anything, the quantitative analyses show to me that the author has not included enough comparisons in the study to be confident that the variation they show is meaningful.

See above; this has been removed.

Reviewer: 3

Line 530: "the presence of dental battery". Actually, non-hadrosaurid hadrosauroids also possess a dental battery. The difference is that this battery becomes more complex in hadrosaurids, having more than 30 tooth positions and with an occlusal plane reaching a maximum of three in width in the dentary dental battery.

Agreed. I have removed this from the text.

Line 540-541: "The holotype and referred dentary likely come from individuals nearing or at osteological maturity...". You are not erecting a new genus or species, thus what holotype are you referring to? "Potentially" new hadrosaurid does not mean it is a new hadrosaurid.

Done. I have modified this to remove the words "holotype and paratype."

Line 607: "The inside of the dentary is perforated by a hole.." Not hole, "foramen" is the word.

Changed to 'canal' to better describe this feature.

Line 737: "resampling". Do you mean Bootstrap?

Yes-this has been changed.

Lines 739-741: It would be more convenient for the reader if you stated in words what characters are those.

These have been added.

Line 776: should be "South America", not "South American".

This is changed in the revised version.

Line 810: not "evinces" but "evidences".

This has been revised.

Figures 2, 4, 5, and 6 are missing. Maybe you forgot to upload them? Or a glitch of the manuscript site of the journal? Anyway, I cannot fully evaluate many of the characters of the osteology without seeing those images.

Yes-it appears the file size was too large so these did not submit in the combined PDF.

Figure S4B: The cladogram shows Atlantohadros, a genus name that does not exist as you are not erecting it (probably a remnant from an earlier version of the manuscript?).

This has been changed.

Appendix D

As I noted in my previous review, I remain unconvinced by the authors arguments for erecting a new taxon here. My view remains largely unchanged upon reading this subsequent revision. This species description is based on highly fragmentary material, and there is little accounting for intra-specific / ontogenetic variability given the paucity of the available data. But, as I also noted in my prior review, the authors have improved their arguments compared to their initial submission, emphasized the hypothesized autapomorphies, noted several other cases of tyrannosauroid taxa being named based on similarly fragmentary materials, and have added at least some note that this description is potentially contingent on finding less fragmentary material to re-assess in the future, so I am not going to insist they refrain from erecting a new taxon. Indeed, I think our disagreement on this point perhaps simply reflects a difference in philosophical approach to science and degree of caution exercised by this author vs others, as evinced by their prior semi-frequent record of naming (or attempting to name) new species under similar conditions (i.e. highly fragmentary materials and a lack of examination of intra-specific variability and ontogeny). In any event, as noted previously, I've made my concerns here noted, and I think that so long as the authors are upfront about the nature of this material, then it's fine to allow them to publish their hypothesis and name this taxon. In the future, as more material is discovered, the issue can be re-visited and critically examined in a more robust way.

I see that the authors have also now removed the problematic quantitative sections. That is probably for the best, given that their analytical results were equivocal at best and working against their arguments at worst. I think that collecting more data and doing some kind of more thorough analyses would ideally have been better than removing them outright, but that would have likely required more data to exist than currently does for their new taxon (which as noted above and previously is sort of the crux of the issue in the paper generally). As it stands, and as I noted in my prior review, the analyses weren't really adding much in the first place so it seems fine to remove them and make it clear that the hypothesis is driven on the qualitative assessments and comparisons.

Agreed. I have decided to remove the new tyrannosauroid name from the paper.

The updated phylogenetic analysis is welcome, as are the additional comparisons discussions, though the very low support values in their trees (noted by the author as likely relating to the incompleteness of the taxon) nicely underscore the concerns noted above and by the other reviewer about the relatively low data quality here and it's potential implications for naming a new taxon vs. describing it as Dryptosauridae of uncertain association / with some features that are potentially distinct from Dryptosaurus

Agreed. See above for my decision to not name the tyrannosaur.

Concerning the hadrosaur, since it is no longer being named, I think some minor changes to the MS text might be warranted, since it is still frequently being referred to as a new taxon and discussed in reference to its holotype, etc. It doesn't really have a holotype since you aren't erecting it as a new taxon. I'd just discuss the material as being from this formation, and potentially new due to the following suite of features (a,b,c), with a note that you are not formally naming it due to the fragmentary nature (and other reasons x,y,z). As well, the hadrosaur is still labelled as 'Atlantohadros' in the Supplementary Information (e.g. Fig S5). Clarifying the text this way is mostly for consistency of the taxonomic hypotheses being presented, and shouldn't diminish the author's discussions of the importance of characterizing the biodiversity of this otherwise poorly characterized time and place.

I appreciate this concern, and have revised the manuscript to remove any reference to a novel taxon of hadrosauroid.

minor note: that font change issue is still present on page 38, lines 855-868 of the pdf.

This has been corrected.

Reviewer: 2

Comments to the Author(s)

The manuscript re-describes a fragmentary theropod metatarsus, erecting a new taxon for the material, and presents new hadrosaurid specimens. Together, these specimens provide information on the

evolution of these clades in the Santonian–Campanian, as well as revealing more of the fauna of Appalachia.

My personal taxonomic philosophy, developed through my own work on very fragmentary theropods, is to be overly cautious and only refer specimens to certain clades or taxa when I can confidently eliminate other potential confounds, like ontogeny, individual variation, or taphonomic distortion. However, I understand that not all palaeontologists approach taxonomy in this way, and that arguments can be made for both sides. Thus, while I personally disagree that the material is sufficient to confidently refer to a tyrannosaur or to erect a new taxon, this is based on my philosophy, and I can provide no evidence against the author's hypothesis. I believe they have done as good a job as can be done to support their assertions with our current knowledge of these dinosaurs. Therefore, I believe the manuscript is fit to publish with some very minor revisions, outlined below:

See above. I agree and have chosen not to erect a new taxon.

Throughout: As this is being submitted to Royal Society Open Science, a UK journal, distances (miles) should be presented in metric units, or at least metric conversions should be presented in parentheses alongside Imperial units. The same is true of words with different spellings, e.g. palaeontology, colour, etc.

Throughout: In-text citations will need to be changed to RSOS's style.

Line 208: the abbreviation C&D Canal should be presented in parentheses after the first use of the full name. However, seeing as this abbreviation is only used twice, maybe it should just be spelled out in full?

Line 292–293: A word is missing at the end of line 292 “characteristic of Peacock et al., 2014;”

Line 486: “Monospecies” should be “Monospecific” or “Monodominant”.

Agreed. All these stylistic issues have been corrected.